# Pushing Forward Pareto Frontiers of Proactive Agents with Behavioral Agentic Optimization

Yihang Yao [1]   Zhepeng Cen [1]   Haohong Lin [1]   Shiqi Liu [1]   Zuxin Liu [2]   Jiacheng Zhu [3]   Zhang-Wei Hong [3,4]
Laixi Shi [5]   Ding Zhao [1]

## Abstract

Proactive large language model (LLM) agents aim to actively plan, query, and interact over multiple turns, enabling efficient task completion beyond passive instruction following and making them essential for real-world, user-centric applications. Agentic reinforcement learning (RL) has recently emerged as a promising solution for training such agents in multi-turn settings, allowing them to learn long-horizon decision-making strategies. However, existing pipelines face a critical challenge in balancing task performance with user engagement, as passive agents cannot efficiently adapt to users' intentions while overuse of human feedback increases the burden on users, which forms a Pareto Frontier between these two objectives. To push forward this frontier, we propose Behavior Agentic Optimization (BAO), an agentic RL framework that enhances and regularizes inter-turn behaviors to improve information-gathering capabilities and suppress inefficient or redundant interactions with users. We evaluate BAO on multiple tasks from the UserRL benchmark suite and demonstrate that it substantially outperforms proactive agentic RL baselines in terms of both higher task performance and lower user efforts, while achieving comparable or even superior performance to commercial LLM agents, highlighting its effectiveness for training proactive, user-centric LLM agents in complex multi-turn scenarios. Our website: https://proactive-agentic-rl.github.io/.

## 1. Introduction

Recent advances in Large Language Models (LLMs) have enabled agents to go beyond passive question answering (Guo et al., 2025; Setlur et al., 2024) and instruction following (Ouyang et al., 2022), toward agentic reasoning that equips agents with abilities to autonomously operate through continual interaction with their environments (Wei et al., 2026; Prabhakar et al., 2025; Ye et al., 2022; Yang et al., 2025b; Ouyang et al., 2025). Proactive agents (Lu et al., 2024) that are allowed to interact with an environment, such as a user, tool, or external system, over multiple turns before offering final answers and actions, have demonstrated great capability and potential in interactive coding (Zhou et al., 2025; Xu et al., 2025; Sun et al., 2025), web-automation (Wei et al., 2025b; Wang et al., 2025a;b), and personalized conversation (Li et al., 2025; Qian et al., 2025b;c). Proactive agents are no longer limited to producing a final answer once all necessary context is provided; instead, they can actively discover task-relevant information through interaction processes with users and then generate answers that faithfully reflect the user's underlying intent and preferences.

To enable such proactive agents, agentic reinforcement learning (agentic RL) emerges as a promising framework. In contrast to prior bandit-style RL for single-turn domains such as math reasoning (Kazemnejad et al., 2024) and code generation (Wei et al., 2025a), agentic RL enables learning in multi-turn tasks, where agents make sequential decisions involving multiple interactions with environments and users (Singh et al., 2025; Jin et al., 2025; Yu et al., 2025; Jiang et al., 2025b; Wang et al., 2025c). Recent studies suggest that agentic RL is well-suited for training proactive agents, as it allows models to learn how to collect information strategically, adapt to feedback, and infer latent user intentions and preferences over the course of an interaction (Wu et al., 2025).

Despite its potential, a key challenge in agentic RL for proactive agents lies in balancing task performance with user engagement. In user-involved tasks, agents may rely on frequent feedback and intervention from users to refine their answers, which can improve final response quality

---

Zuxin Liu contributed to this work while at Salesforce AI Research. [1]Carnegie Mellon University [2]Salesforce AI Research [3]Massachusetts Institute of Technology [4]MIT-IBM Watson AI Lab [5]Johns Hopkins University. Correspondence to: Yihang Yao <yihangya@andrew.cmu.edu>.

and help reconcile mismatched intentions (Lu et al., 2024). Guided by task rewards, RL agents can exploit this mechanism with extensive interaction. However, this comes at a cost: repeated or redundant requests for human engagement can erode user confidence in the agent's competence. This creates a fundamental trade-off between the quality of final completions and the level of user intervention required: with more interactions with users, agents gather more information and improve their performance, yet the user's satisfaction decreases due to the tedious interactions.

In this work, we formulate this challenge as a multi-objective optimization (MOO) problem, where user engagement and task performance constitute potentially conflicting objectives and form a Pareto frontier. In contrast to concurrent works that introduce handcrafted reward signals to balance these objectives (Sun et al., 2025; Wu et al., 2025), we emphasize the importance of *inter-turn behaviors*. We propose BAO, which not only optimizes agents with respect to task rewards, but also enhances and regularizes inter-turn behaviors, including retrospective reasoning and prospective planning, to enable more effective exploration and exploitation of task-relevant information. The empirical results further show that our method efficiently pushes forward the Pareto frontier between user engagement and task performance compared to the agentic RL baseline for proactive agents. Our key contributions are summarized as:

- This work formulates proactive agent training as an MOO problem and identifies the Pareto frontier between final task performance and user engagement efforts.
- We characterize multi-turn behaviors, including retrospective reasoning and prospective planning, that connect actions with historical information and future scheduling.
- We propose BAO, which enhances and regularizes multi-turn behaviors in proactive agent training to collect and utilize contextual feedback more effectively.
- Extensive experiments demonstrate that BAO significantly improves task performance without overusing user engagement, surpassing strong RL baselines and commercial models on the UserRL benchmark.

## 2. Related Works

**Proactive agents** have been proposed to study how agents interact with environments and adapt to users' feedback, objectives, and intentions (Sun et al., 2025). They are also discussed in personalization (Zhao et al., 2025; Zhu et al., 2025; Li et al., 2025; Abdulhai et al., 2025; Jiang et al., 2025a; Shenfeld et al., 2025a) and task-oriented dialogue systems (Mo et al., 2024), expanding their capabilities from passive response generation to anticipatory assistance (Ye et al., 2022). Recent efforts have focused on enhancing agent autonomy through integrating information retrieval and sophisticated planning for sequential action (Dong et al.,

2025). CollabLLM (Wu et al., 2025) utilizes multi-turn aware reward considering token efficiency and task completion, and finetunes agents for better human-AI collaboration. Beyond standard task completion, recent literature increasingly leverages RL to explicitly formulate and optimize for user-centric traits such as proactivity and personalization. Sun et al. (2025) measures proactivity by the minimization of user effort and personalization by adherence to user preferences and leverages multi-objective RL to balance them. To address the complexities of multi-turn optimization, Qian et al. (2025c) introduces various approaches of advantage estimation, while Zhou et al. (2025) proposes constructing turn-wise preference pairs to facilitate DPO (Rafailov et al., 2023). Despite these advances, how to efficiently manage the trade-off between user engagement and task performance remains largely unexplored. In addition, while some concurrent works discuss using RL to train proactive agents (Sun et al., 2025), a systematic study of the impact of multi-turn *behaviors* on proactive agents remains underexplored.

**Reinforcement learning** has emerged as a promising training paradigm for LLMs, enabling stronger reasoning capabilities (Jaech et al., 2024; Guo et al., 2025). It has demonstrated effectiveness in training specialized models across a range of domains, including mathematical reasoning (Shen et al., 2025; Qu et al., 2025; Shenfeld et al., 2025b; Kang et al., 2025) and code generation (Zeng et al., 2025a). Beyond single-turn tasks, RL has recently been extended to multi-turn agentic settings (Qian et al., 2025a; Zeng et al., 2025b; Ning et al., 2025; Liu et al., 2025b;a; Paglieri et al., 2024; Chen et al., 2025; Lu et al., 2025; Luo et al., 2025). Prior work has studied challenges such as credit assignment (Zeng et al., 2025b) and efficient exploration (Wan et al., 2025) in multi-turn RL. In parallel, research on thinking token patterns has analyzed long chain-of-thought reasoning (Yeo et al., 2025) and identified behaviors such as self-verification and reflection that improve RL efficiency (Gandhi et al., 2025; Cen et al., 2025; Yao et al., 2025). However, these studies focus on single-turn settings. Inter-turn behaviors in agentic reasoning remain largely underexplored, especially for proactive multi-turn interaction.

## 3. Problem Formulation

**Contextual MDP.** We formulate the interaction of a proactive LLM agent as a finite-horizon Contextual Markov decision process (Contextual MDP) (Hallak et al., 2015), defined by the tuple $\mathcal{M} = \{\mathcal{S}, \mathcal{A}, P, r, c, \mu_0\}$, where $\mathcal{S}$ and $\mathcal{A}$ denote the state and action spaces, and $\mu_0$ denotes the initial state distribution. We denote $c$ as a hidden context to explicitly model the user's characteristics/preferences, which is unobservable to the agent. The context

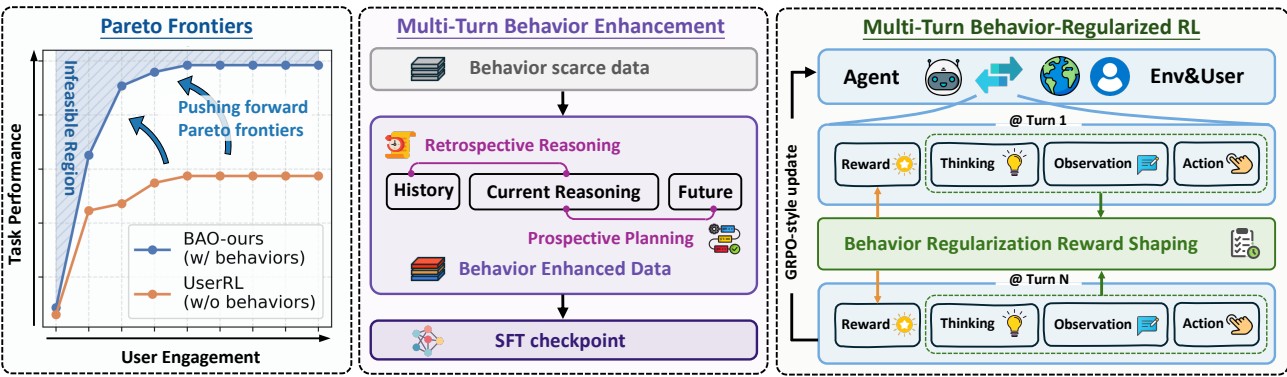

*Figure 1.* Pareto-Frontier and the BAO pipeline. (**Left**): Pareto-Frontiers between user engagement efforts and task performance. The agent should maximize performance while minimizing unnecessary user interaction, and we aim at minimizing the "infeasible region". (**Middle**): Behavior Enhancement. (**Right**): Behavior-Regularized RL.

$c$ remains fixed throughout the whole episode. At turn $t$, the agent observes a state $s_t \in \mathcal{S}$ and generates an action $a_t \in \mathcal{A}$, corresponding either to interacting with the environment to acquire information or to responding to the user with an answer based on the interaction history $h_t = (s_1, a_1, \ldots, s_t)$. The underlying context $c$ governs both transition dynamics $s_{t+1} \sim P(\cdot \mid s_t, a_t, c)$ and reward function $r : \mathcal{S} \times \mathcal{A} \times \mathcal{S} \times c \rightarrow \mathbb{R}$. The agent operates under a fixed interaction budget $T$. Let $\tau = \{(s_t, a_t, r_t)\}_{t=1}^{t_{\text{end}}}$ denote the interaction trajectory terminated at $t_{\text{end}} \leq T$.

**Pareto Frontiers of Proactive Agents.** The objective of proactive agents is to improve task performance through interaction with users while minimizing unnecessary user efforts. This forms a natural trade-off and Pareto frontier (Deb, 2011) between these two objectives. We then formalize this problem based on the contextual MDP formulation. Denote the proactive agent policy as $\pi_\theta(a_t \mid h_t)$, which maps the interaction history to actions. We consider a structured action space $\mathcal{A} := \mathcal{A}_u \cup \mathcal{A}_e$ consisting of two subspaces: 1) *User involved action space* $\mathcal{A}_u$: includes actions that require explicit user feedback, such as answer verifications. 2) *Environment involved action space* $\mathcal{A}_e$: includes actions that interact with tools or external systems without user intervention[1].

Let the accumulated reward $R(\tau) := \sum_t r_t$ denote **task performance**, and let $U(\tau) = |\{t : a_t \in \mathcal{A}_u\}|$ denote **user engagement**, measured as the number of user-involved actions per trajectory. These two objectives define a Pareto frontier, as illustrated in Figure 1 (**Left**). Our goal is to push this frontier by reducing the infeasible region.

**Challenge: Naive Reward Balancing Fails.** A straightforward method to balance the trade-off is to introduce a

---

[1]A detailed explanation of action types in our user-centric tasks is provided in Appendix B.1.

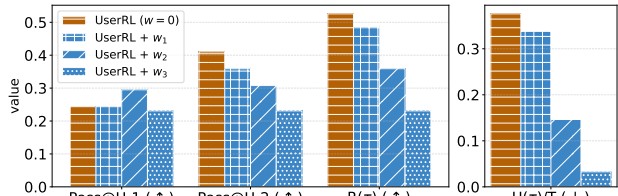

*Figure 2.* The performances on two objectives with different penalty weights $w_1 < w_2 < w_3$ in (1). Pass@U-$k$ is defined as the pass rate when allowing up to $k$ User-involved actions per trajectory ($U(\tau) = k$). ↑, ↓: The higher/lower, the better. Simply tuning weights fails to improve the trade-off between task performance maximization and use engagement minimization.

user engagement reward in policy optimization:

$$\max_{\pi_\theta} \quad \mathbb{E}_{\tau \sim \pi_\theta} \big[ R(\tau) - w\, U(\tau) \big] \quad \text{s.t.} \quad |\tau| \leq T, \quad (1)$$

where $w$ is a tunable weight that controls the trade-off between task performance and the burden of user involvement in the entire process. Ideally, the learning objective in (1) encourages policies to balance task performance and user-engagement effort, eliciting user interaction only when necessary to improve performance. However, this naive approach fails in practice: penalizing the number of interactions can reduce user effort, but may also harm task performance by limiting proactive exploration, as shown in Figure 2. As a result, instead of advancing the Pareto frontier, it often leads to suboptimal trade-offs that deviate from our goal. One key reason for this failure is that sparse reward signals are insufficient to effectively guide learning. This highlights the need to model *inter-turn behaviors*, which connect an agent's current reasoning and actions with past information and future planning.

## 4. Method

We have shown that simply tuning the weight $w$ between the two objectives in (1) fails to simultaneously improve

task performance and minimize user engagement. A key reason is the lack of efficient inter-turn proactive behaviors: agents fail to learn effective strategies for leveraging past information and planning future actions in user-centric tasks. To address this issue, we set $w = 0$ in (1) and instead focus on *agentic behaviors* that enhance reasoning capabilities, enabling more effective optimization of both objectives.

In this section, we first introduce multi-turn proactive behaviors that explicitly connect current decisions with historical information and future planning for efficient information gathering and task solving. Building on these behaviors, we then present Behavioral Agentic Optimization (**BAO**), a behavior-integrated framework for agentic RL.

### 4.1. Multi-Turn Behaviors for Proactive Agents

As shown in Figure 1, we introduce two classes of multi-turn behaviors: **(1) Retrospective Reasoning**, which integrates and revises information from history, and **(2) Prospective Planning**, which bridges current reasoning to future actions.

#### 4.1.1. RETROSPECTIVE REASONING BEHAVIORS

The retrospective reasoning behaviors focus on integrating and revising information from history to facilitate agentic reasoning. We consider the following two patterns:

**Memory Management.** The memory is used to maintain and update hypotheses about the hidden context $c$. With the memory, the agent retrieves and links related information from history $h_t$ to inform current decision making, which enables consistent reasoning across turns and prevents redundant or contradictory actions.

**Hypothesis Refinement.** When new observations contradict existing hypotheses, the agent should refine structured hypothesis instead of falling into unproductive guessing or repetition loops. This behavior encourages the agent to explicitly revise incorrect assumptions, identify the source of failure, and adjust its subsequent actions accordingly, leading to more robust recovery from early mistakes.

#### 4.1.2. PROSPECTIVE PLANNING BEHAVIORS

Planning for the future is another important agentic feature, which we refer to as prospective planning behaviors. We also consider two types of behaviors:

**Dynamic Scheduling.** The agent adapts its strategy according to interaction budgets. In the early stage of the interaction, the agent prioritizes information gathering, while later turns increasingly favor consolidation and answer submission. This budget-aware planning helps balance exploration and exploitation within finite-horizon constraints.

**Strategical Querying.** Reducing uncertainty about the underlying condition is another key axis. The agent should

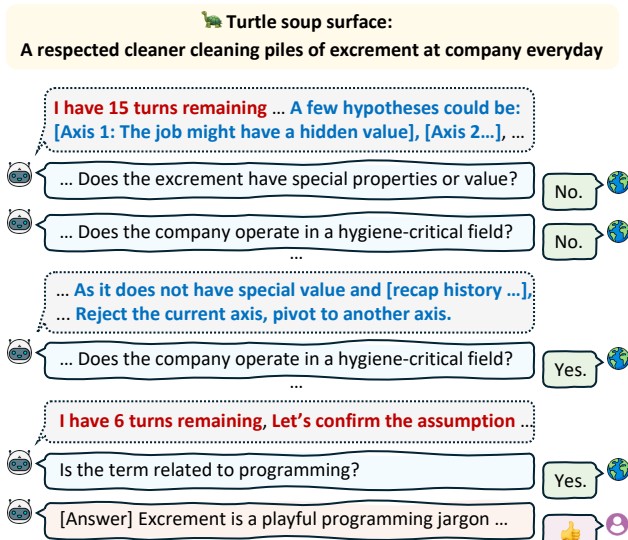

*Figure 3.* Behavior examples from Turtle-Gym. Hidden twist: This person is a programmer; in the computer industry, old, large, and difficult-to-maintain code is referred to as a *pile of excrement*. **Red**: Prospective Planning; **Blue**: Retrospective Reasoning.

actively seek task-relevant information through tool calls and interactions, while avoiding vague or repeated requests for similar information.

### 4.2. Behavioral Agentic Optimization

As shown in Figure 1, our BAO has two key components: (i) behavior enhancement, which enforces inter-turn proactive behaviors; and (ii) behavior-regularized RL, which shapes agentic behaviors during policy update.

#### 4.2.1. BEHAVIOR ENHANCEMENT

To incorporate these multi-turn behaviors into the student LLMs, we adopt a warm-start training pipeline (Gandhi et al., 2025), where we first perform SFT on pre-trained models to familiarize them with domains and inject the multi-turn behaviors, and then use RL to further incentivize agentic reasoning capabilities. We synthesize the SFT dataset by explicitly prompting an external teacher model to generate sample data with the specified behaviors. In practice, we use GPT-4o as the teacher model. The prompt and more details are provided in Appendix C.

We present behavior examples from Turtle-Gym, an experimental task in which the agent queries the environment for clarification to uncover a hidden twist, simulating a challenging proactive interaction task for discovering implicit information. For clarity, we visualize representative interaction traces in Figure 3. For prospective planning, the agent tracks the remaining interaction budget and guides action selection accordingly. For example, it initializes an assumption set at the beginning and verifies these assumptions

before answer submission when sufficient budget remains. For retrospective reasoning, the agent maintains and recursively updates an assumption set based on environmental feedback, using interaction history to decide whether to continue along the current reasoning axis or pivot to a new one. Full interaction traces and additional details are provided in Appendix D.1.

### 4.2.2. BEHAVIOR-REGULARIZED RL

While the retrospective and planning behaviors effectively facilitate the multi-turn reasoning, we observe that they also introduce undesired behaviors. Therefore, we regularize them via *turn-level reward shaping* in RL, which augments feedback rewards with penalties to calibrate the agent's interaction dynamics. The regularizations include:

**Information-Seeking Regularization.** One common failure mode of proactive agents is inefficient information-seeking in prospective planning, where the agent repeatedly requests user interactions without information gain from the environment. To encourage comprehensive evidence gathering before assumption revision and reduce overuse of human engagement, we penalize consecutive request of user interactions without information gain from the environment:

$$r_t \leftarrow r_t - \lambda_{\text{ans}}, \text{ if } a_t, a_{t-1} \in \mathcal{A}_u, \quad (2)$$

where $\lambda_{\text{ans}} > 0$ controls the penalty scale.

**Over-Thinking Regularization.** Another common failure of proactive agents is premature exhaustion of the token budget caused by excessive thinking in retrospective reasoning. To prevent this, we penalize trajectories that fail to interact sufficiently with the environment or user before termination. Specifically, if the agent fails to complete the task and the actual trajectory contains only $T'$ turns ($T' < T$), we introduce a penalty to the reward of each turn:

$$r_t \leftarrow r_t - \lambda_{\text{think}}(T - T')/T', \quad (3)$$

where $\lambda_{\text{think}}$ is the penalty coefficient. In implementation, we add an additional penalty $r_t \leftarrow r_t - \lambda_{\text{think-0}}$ where $t = 1, \lambda_{\text{think-0}} = 0.5$ for cases that fail to complete initial-turn generation within the token budget.

With the above regularizations, we run RL on the SFT-trained models to further optimize the policy while eliminating the undesired interaction patterns. In practice, we adopt GRPO (Shao et al., 2024) for policy update. At each training iteration, for each task, we sample a group of $N$ trajectories $\{\tau_i\}_{i=1}^N$ from the current policy $\pi_\theta$. Each trajectory $\tau_i = \{(s_t^i, a_t^i, r_t^i)\}_{t=1}^{T_i}$ consists of multi-turn interactions with turn-level rewards. The GRPO objective maximizes the clipped surrogate loss:

$$\mathcal{L} = \mathbb{E}_{i,t} \left[ \min \left( \rho_t^i \hat{A}_t^i, \text{ clip}(\rho_t^i, 1-\epsilon, 1+\epsilon) \hat{A}_t^i \right) \right], \quad (4)$$

---

**Algorithm 1** BAO Training Pipeline

**Input:** Seed dataset $\mathcal{D}_{\text{seed}}$; RL tasks $\mathcal{D}_{\text{RL}}$; policy $\pi_\theta$; teacher $\pi_T$.
**Output:** Trained proactive policy $\pi_{\theta'}$.
 1: # Behavior-Enhanced SFT
 2: Initialize $\mathcal{D}_{\text{SFT}} \leftarrow \emptyset$
 3: **for** each instance $x \in \mathcal{D}_{\text{seed}}$ **do**
 4:     Construct behavior prompt $p_{\text{BE}}$
 5:     Generate enhanced trace $\tau^{\text{BE}} \leftarrow \pi_T(\cdot \mid x, p_{\text{BE}})$
 6:     $\mathcal{D}_{\text{SFT}} \leftarrow \mathcal{D}_{\text{SFT}} \cup \{(x, \tau^{\text{BE}})\}$
 7: **end for**
 8: Supervised finetune: $\pi_\theta \leftarrow \text{SFT}(\pi_\theta, \mathcal{D}_{\text{SFT}})$
 9: # Behavior-Regularized RL
10: **repeat**
11:     Rollout $\{\tau_i\}_{i=1}^N \sim \pi_\theta(\cdot \mid s_0), \ s_0 \sim \mathcal{D}_{\text{RL}}$
12:     Apply behavior regularization (2, 3)
13:     Update policy with GRPO objective (4, 5)
14: **until** convergence
15: **Return:** $\pi_{\theta'}$

---

where $\rho_t^i$ is the policy likelihood ratio, $\hat{A}_t^i$ is the advantage estimate, and $\epsilon$ is the clipping threshold.

Our implementation computes turn-level advantage (Zeng et al., 2025b; Qian et al., 2025c), where each turn receives a scalar credit and broadcasts it to all tokens within that turn. Suppose the trajectory $\tau_i$ has $T_i$ turns, with regularized turn rewards $\{r_t^i\}_{t=1}^{T_i}$ obtained. With discount $\gamma \in (0, 1]$, we compute reward-to-go $G_t^i$, trajectory total return $R^i$, and advantages $\hat{A}_t^i$:

$$G_t^i = \sum_{j=k}^{T_i} \gamma^{j-k} r_j^i, \ R^i = \sum_{k=1}^{T_i} r_k^i, \quad (5a)$$

$$\hat{A}_t^i = \frac{G_t^i - \text{mean}(\{R^i\}_{i=1}^N)}{\text{std}(\{R^i\}_{i=1}^N)}. \quad (5b)$$

## 5. Experiments

### 5.1. Experiment Setup

**Tasks.** We perform experiment evaluation on three proactive agent tasks from the UserRL benchmark (Qian et al., 2025c): **Function-Gym**, **Telepathy-Gym**, and **Turtle-Gym**. These tasks allow up to $T = 15$ interaction turns in one trajectory. The actions of these tasks involve interacting with the environment to collect information, submitting answers to users, and receiving feedback. Function-Gym uses fully rule-based feedback and rewards, resulting in minimal sim-to-real gap and enabling controlled analysis of RL optimization. Telepathy-Gym introduces feedback distribution shift: feedback is generated by Qwen3-8B (Yang et al., 2025a) during training, while GPT-4o (Hurst et al., 2024) is used as the user simulator at evaluation. Turtle-Gym further

*Table 1.* Proactive tasks experiments. Pass@U-$k$ is defined as the pass rate when allowing up to $k$ U̲ser-involved answer actions within a single interaction trajectory (when $U(\tau) = k$). Score means unshaped cumulative reward. "w/o BE" and "w/o BR" indicate the ablations without behavior enhancement and reward regularization, respectively. **Bold**: Finetuned agents with highest Pass@U-1 or Pass@U-2. Blue: Finetuned agents with lowest user-involved rate (UR). ↑, ↓: The higher/lower, the better.

| | Function-Gym | | | | Telepathy-Gym | | | | Turtle-Gym | | | |
|---|---|---|---|---|---|---|---|---|---|---|---|---|
| | Pass@U-1(↑) | Pass@U-2(↑) | Score(↑) | UR(↓) | Pass@U-1(↑) | Pass@U-2(↑) | Score(↑) | UR(↓) | Pass@U-1(↑) | Pass@U-2(↑) | Score(↑) | UR(↓) |
| *Commercial Models* | | | | | | | | | | | | |
| Gemini-3-flash | 0.4103 | 0.4231 | 0.4231 | 0.0440 | 0.8049 | 0.8293 | 0.8537 | 0.1414 | 0.1823 | 0.2073 | 0.2073 | 0.0613 |
| Gemini-2.5-Pro | 0.3205 | 0.4359 | 0.4359 | 0.1003 | 0.5854 | 0.6585 | 0.7317 | 0.2330 | 0.3156 | 0.3469 | 0.3844 | 0.2250 |
| Gemini-2.5-Flash | 0.2308 | 0.3333 | 0.3718 | 0.1290 | 0.5853 | 0.6585 | 0.7073 | 0.1625 | 0.1604 | 0.1844 | 0.2198 | 0.2155 |
| GPT-5-0807 | 0.5000 | 0.6923 | 0.7436 | 0.1365 | 0.7317 | 0.8293 | 0.8293 | 0.1270 | 0.3500 | 0.3521 | 0.3521 | 0.0897 |
| GPT-4o | 0.1282 | 0.1282 | 0.1410 | 0.2248 | 0.4634 | 0.5854 | 0.6342 | 0.1747 | 0.1719 | 0.2333 | 0.2750 | 0.2553 |
| GPT-4o-mini | 0.0256 | 0.0256 | 0.0385 | 0.1767 | 0.4634 | 0.5610 | 0.6216 | 0.1183 | 0.0598 | 0.0732 | 0.0793 | 0.3323 |
| *Open-Source (Raw Models)* | | | | | | | | | | | | |
| Qwen3-32B (Raw) | 0.1667 | 0.1795 | 0.1795 | 0.0379 | 0.5366 | 0.5610 | 0.5854 | 0.1913 | 0.1000 | 0.1500 | 0.1688 | 0.3082 |
| Qwen3-14B (Raw) | 0.1154 | 0.1154 | 0.1282 | 0.0389 | 0.3902 | 0.4634 | 0.4634 | 0.2532 | 0.1281 | 0.1406 | 0.1500 | 0.5264 |
| Qwen3-8B (Raw) | 0.0513 | 0.0513 | 0.0513 | 0.0559 | 0.3902 | 0.4390 | 0.4878 | 0.1699 | 0.0792 | 0.0938 | 0.1135 | 0.4778 |
| *Fine-tuned Models* | | | | | | | | | | | | |
| UserRL-4B | 0.2436 | 0.4103 | 0.5256 | 0.3758 | 0.4878 | 0.5122 | 0.6585 | 0.4251 | 0.0417 | 0.0563 | 0.0594 | 0.7617 |
| BAO (ours)-4B | 0.2692 | 0.5354 | 0.6923 | 0.2148 | 0.5123 | 0.6585 | 0.6585 | 0.1870 | 0.1063 | 0.1125 | 0.1125 | 0.2917 |
| BAO (ours) w/o BE-4B | 0.2436 | 0.3718 | 0.4744 | 0.2483 | 0.4878 | 0.5366 | 0.5610 | 0.1452 | 0.0802 | 0.0979 | 0.1146 | 0.3359 |
| BAO (ours) w/o BR-4B | 0.3333 | 0.4615 | 0.7179 | 0.3755 | 0.3902 | 0.4390 | 0.5610 | 0.5290 | 0.0729 | 0.0750 | 0.0813 | 0.8698 |
| UserRL-1.7B | 0.1538 | 0.2692 | 0.2949 | 0.3193 | 0.2683 | 0.2683 | 0.4390 | 0.6543 | 0.0479 | 0.0510 | 0.0583 | 0.7721 |
| BAO (ours)-1.7B | 0.1923 | 0.3205 | 0.3590 | 0.2133 | 0.5610 | 0.5854 | 0.6097 | 0.1601 | 0.0563 | 0.0667 | 0.0667 | 0.2976 |
| BAO (ours) w/o BE-1.7B | 0.1410 | 0.2179 | 0.2436 | 0.2404 | 0.4390 | 0.4878 | 0.4878 | 0.1559 | 0.0615 | 0.0660 | 0.0750 | 0.3776 |
| BAO (ours) w/o BR-1.7B | 0.2308 | 0.2949 | 0.3974 | 0.4284 | 0.1707 | 0.1951 | 0.4390 | 0.6911 | 0.0594 | 0.0625 | 0.0771 | 0.9064 |

increases difficulty with narrative reasoning, using Qwen3-8B for both feedback and reward modeling during training and GPT-4o for evaluation, inducing distribution shift in both feedback and rewards. In the UserRL benchmark, actions are unified into three types: `Action`, `Search`, and `Answer`. The first two belong to the *environment-involved action space* $\mathcal{A}_e$, which the agent uses to interact with the environment. The last one belongs to the *user-involved action space* $\mathcal{A}_u$, which the agent uses to submit an answer and receive feedback from the user.

**Training and Evaluation.** We use Qwen3 series models as base models and vary the model parameter with $1.7$ B and 4B. The training and evaluation datasets are both from the original UserRL benchmark. In the main experiments, in addition to the trajectory-wise unshaped accumulative reward $\mathbb{E}[R(\tau)]$, we also report Pass@U-$\{k\}$, defined as the pass rate when allowing up to $k$ U̲ser-involved answer submissions within a single interaction trajectory. This metric captures an agent's ability to iteratively refine its reasoning and recover from early mistakes under a fixed interaction budget. We further report the User Involvement Rate (UR), defined as the proportion of user-involved actions among all action types in a trajectory $\mathbb{E}[U(\tau)/|\tau|]$; a lower UR indicates fewer user engagements, reflecting reduced human burden and correspondingly higher user satisfaction. For fair comparison, we report the best results of UserRL (Qian et al., 2025c) among several configurations provided in their paper. More details, including the task and action types for $\mathcal{A}_u, \mathcal{A}_e$ descriptions, are available in Appendix B.

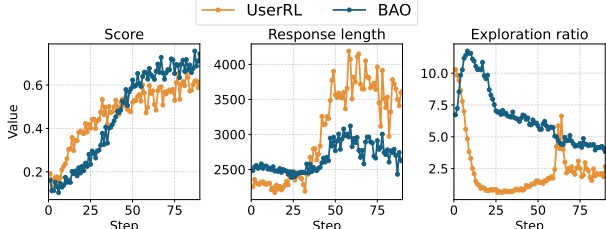

*Figure 4.* Function-Gym training curves. BAO keeps a higher exploration ratio, achieving higher task performance with even fewer generated tokens compared to UserRL.

## 5.2. Main Results

The results of the main experiment and ablation studies are presented in Table 1. We compare our trained models with closed-source models from the Gemini (Team et al., 2023) and GPT (Hurst et al., 2024) series, raw base models from the Qwen3 series, and a baseline proactive RL method, UserRL (Qian et al., 2025c). For all RL fine-tuned models, we ensure a fair comparison by using the same number of SFT and RL training samples and training epochs.

As shown in Table 1, even frontier commercial models fail to achieve satisfactory performance on proactive agent tasks such as Function-Gym and Turtle-Gym. For example, on Function-Gym, Gemini-3-Flash attains a high Pass@U-1 but does not benefit from additional user interaction budgets, resulting in a low task score. The RL-based method UserRL improves the performance of base models by training on demonstrations and through interaction-based learning. However, UserRL still faces several challenges. First, it exhibits a high action rate, indicating that it relies heavily

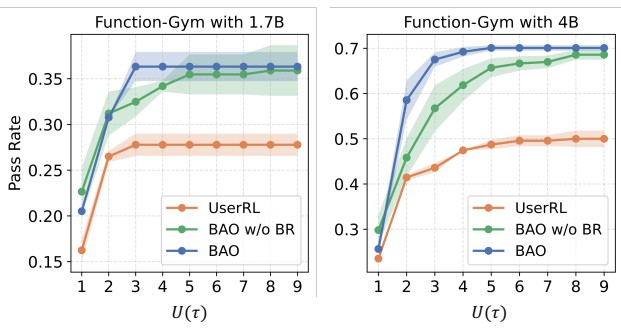

*Figure 5.* Pareto frontiers in Function-Gym. The results are averaged over three random seeds. The shaded area represents the standard deviation. BAO is with better Pareto Frontiers.

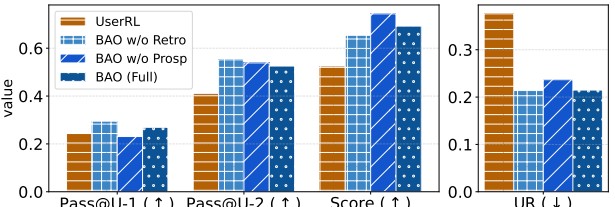

*Figure 6.* Behavior analysis on Function-Gym with Qwen3-4B as base models. $\uparrow, \downarrow$: the higher/lower, the better. Full behaviors enable a more balanced performance in BAO.

on user engagement to verify answer correctness rather than autonomously exploring the environment to gather information. Second, it shows a low Pass@U-$k$ rate. This suggests that it often fails to confidently produce correct answers in the first several submissions, potentially reducing users' trust in the trained models. We also provide the failure case analysis in Appendix A.4 where UserRL fails due to poor inter-turn capabilities, with inefficient retrospective and planning behaviors.

In contrast, our BAO achieves better performance, with generally higher Pass@U-$1, 2$ values and a significantly lower action rate. This indicates that the trained model has greater confidence in producing correct answers at the early answer submissions and relies less on user engagement for feedback and interventions. In addition, the learned models demonstrate comparable or even superior performance relative to closed-source frontier models. The advantage is particularly evident on Function-Gym tasks. For example, the 1.7B model surpasses the performance of GPT-4o-mini, and the 4B checkpoint outperforms GPT-4o in terms of score.

The training curves on Function-Gym are visualized in Figure 4, where BAO converges to a higher score during RL compared to UserRL. Meanwhile, during the training, the average response length increases slightly while remaining within a reasonable range. In contrast, UserRL exhibits a dramatic increase in response length with only marginal performance gains, suggesting potential overthinking. We also visualize the exploration rate, defined as the ratio between information-seeking actions and answer submission actions $\mathbb{E}[|\{a_t \in \mathcal{A}_e\}|/|\{a_t \in \mathcal{A}_u\}|]$. BAO maintains a higher level of exploration for information compared to the baseline, which helps explain its stronger overall performance.

We further visualize the Pareto frontiers of the learned models in Figure 5. The x-axis represents the number of allowed user-involved actions, and the y-axis indicates the pass rate, where the upper-left region corresponds to the Pareto-optimal frontier. From these results, we can clearly

observe that our BAO pushes the frontier forward compared to the baseline RL method UserRL, indicating BAO can better gather information, utilize human feedback, and make correct inferences on the hidden context.

We also conduct ablation studies on the choice of teacher model for behavior enhancement in SFT, as well as on the hyperparameters $\lambda_{\text{ans}}$ and $\lambda_{\text{think}}$, as detailed in Appendix A.2 and A.3. The results show that our BAO maintains stable performance under a 10× variation of $\lambda_{\text{ans}}$ and $\lambda_{\text{think}}$, and continues to outperform the RL baseline UserRL when switching to an alternative teacher model.

> **Main Experiment Takeaways**
> - Prior RL baseline models struggle with proactive agent tasks, with low Pass@U-$k$ and high UR.
> - BAO learns more efficient proactive behaviors.
> - BAO pushes forward the Pareto frontiers of agents and approaches or exceeds commercial models.

### 5.3. Behavior Ablations

We conduct ablation studies on two key components, behavior enhancement and behavior regularization, by removing each component from the full BAO. The results in Table 1 show that removing behavior regularization leads to a large increase in the answer rate, indicating that the learned models tend to rely more heavily on user verification and feedback on final answers to achieve high scores after RL. Figure 5 further illustrates the role of behavior regularization in pushing the performance boundaries. In addition, behavior regularization alone is insufficient. When combined with behavior enhancement, the model achieves better overall performance, with generally higher Pass@U-$k$ values and scores, as shown in Table 1.

To investigate the effectiveness of the *Retrospective Reasoning* and *Prospective Planning* behaviors introduced in Section 4.1, we conduct an ablation study by controlling the enhanced behavior types included in the SFT dataset. The results are shown in Figure 6. We observe that *Retrospective Reasoning* increases the upper bound of the pass rate, as

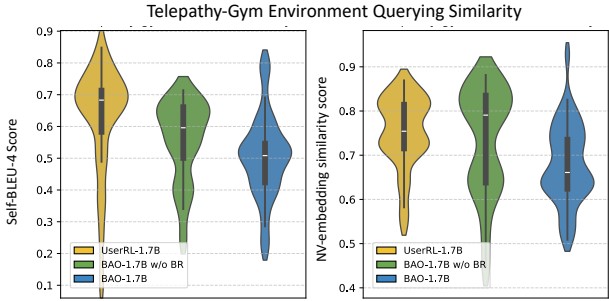

Figure 7. Information querying diversity evaluation in Telepathy-Gym. BAO queries more diverse information during interactions.

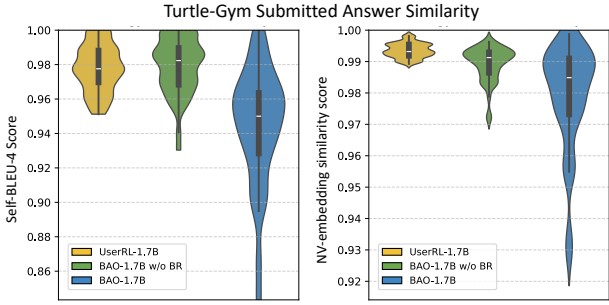

Figure 8. Answer diversity evaluation in Turtle-Gym. BAO explores answer space better by generating more diverse answers.

reflected by the highest *Score* value, which corresponds to the pass rate achievable when the agent is allowed to submit answers until the interaction budget is exhausted. However, retrospective reasoning alone does not necessarily improve Pass@U-1. One straightforward explanation is that only when the number increases does the incorporation of history start to show its capability. In contrast, *Prospective Planning* improves Pass@U-1 due to its planning-ahead capability. Nevertheless, prospective planning does not achieve the highest Score, indicating that under longer interaction horizons, retrospective reasoning is still required to effectively summarize past information and connect it to future actions. By combining these two behaviors, we achieve balanced performance in both metrics, as demonstrated by the full BAO configuration.

> **Behavior Ablations Takeaways**
>
> • Behavior Regularization is critical to encouraging exploration and reducing reliance on user verification, and it is most effective when combined with Behavior Enhancement.
> • Retrospective reasoning and prospective planning are complementary, where their combination yields balanced improvements in both early accuracy and long-horizon performance.

### 5.4. Exploration Evaluation

The performance improvement of BAO largely stems from its enhanced exploration capability, in both information gathering and inferring hidden context. In addition to the exploration metrics, the exploration ratio, presented in Figure 4, we further provide a quantitative evaluation from both lexical and semantic perspectives. Specifically, we use Self-BLEU scores and embedding similarity scores computed with the NV-Embed-v2 model (Lee et al., 2024) to assess diversity in information querying and answer space exploration. The information-gathering results on Telepathy-Gym and the answer-space exploration results on Turtle-Gym are shown in Figures 7 and 8, respectively. For both metrics,

lower similarity scores indicate greater diversity within an interaction trajectory and stronger exploration capability.

From Figure 7, we observe that compared to UserRL, our BAO significantly increases diversity during the information-gathering stage, as reflected by lower lexical and semantic similarity scores. Results in Figure 8 further indicate that after information gathering, BAO explores the answer space more efficiently by reducing answer redundancy when submitting answers following failures. Another finding is that relying solely on multi-turn behavior enhancement during SFT is insufficient to preserve these desirable exploration behaviors; combining it with regularization during RL yields the strongest capabilities.

> **Exploration Experiments Takeaway**
>
> • BAO substantially enhances exploration in both information gathering and answer space exploration, leading to more efficient and less redundant interactions with environments and users.

### 5.5. Reward Hacking Discussion

When using LLM-as-judges in RL, reward hacking can arise (Zhai et al., 2023; Miao et al., 2024). In our experiments, we also observe this pattern most prominently in Turtle-Gym, where the sim-to-real gap is most substantial: during training, we use Qwen3-8B as the user simulator and reward model, whereas testing is conducted with GPT-4o. A common form of reward hacking of this task involves repeatedly generating long answers to confuse the judge model, inducing it to leak hidden knowledge or assign positive rewards even when the response does not explicitly satisfy the target rubrics.

To quantitatively characterize this issue, we visualize the average number of answer tokens per trajectory, the reward score, and pass@U-1 in training and testing in Figure 9. We observe that UserRL produces significantly longer answers and achieves higher training scores and pass@U-1. However, during evaluation, both the score and pass@U-1

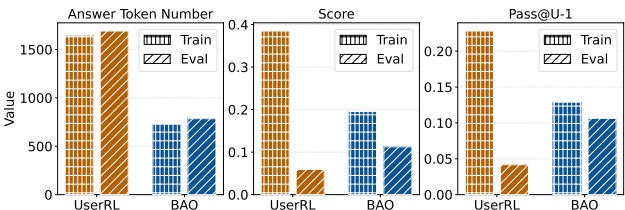

*Figure 9.* Turtle-Gym reward hacking issue analysis with Qwen3-4B as base models. BAO achieves a lower occurrence of reward hacking, leading to a higher evaluation score.

drop sharply, despite a high reward translation rate (RTR = 0.154), defined as the ratio between evaluation and training rewards. In contrast, although our BAO attains relatively lower scores and pass@U-1 during training, it consistently outperforms UserRL at evaluation time while maintaining a comparably high RTR = 0.575. This indicates that BAO effectively mitigates reward hacking by encouraging desirable interaction behaviors and applying behavior regularization during RL training.

> **Reward Hacking Takeaway**
>
> • BAO can reduce the reward hacking issue when using LLM-as-Judges.

## 6. Conclusion

In this work, we study proactive agent training through the lens of agentic RL and identify a fundamental trade-off between task performance and user engagement in interactive settings. By formulating this challenge as an MOO problem, we show that effective proactive agents must balance intention discovery quality with interaction efficiency. To address this challenge, we introduce BAO, which combines behavior enhancement and behavior regularization to shape turn-level agentic reasoning across multi-turn interactions. Among these behaviors, we identify retrospective reasoning and prospective planning as key components that connect current decisions with past interaction history and future planning, and that play a central role in proactive agent training. Empirical results across diverse proactive agent tasks demonstrate that BAO consistently pushes the Pareto frontier forward, achieving stronger task outcomes while maintaining high user satisfaction. Our ablation studies and analyses further highlight the importance of explicitly modeling engagement-aware objectives in agentic RL, and suggest a principled direction for building reliable and user-aligned proactive agents.

One limitation of our current work is that it focuses on language-only agents; future work will extend the proposed pipeline to multimodal agent training with broader application domains. Nevertheless, we hope that our findings provide insights into addressing the trade-off between user engagement and task performance in proactive agents.

## Acknowledgment

We would like to thank the anonymous reviewers and the decision committee for their review efforts and constructive feedback.

## Impact Statement

This paper presents work whose goal is to advance the field of Machine Learning. One potential negative social impact of this paper is the misuse of our method, and unsafe multi-turn behaviors can lead to harmful consequences for proactive LLM agents.

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

# A. Additional Experiments

Due to the space limit, we moved some experiments from the main context here.

## A.1. Supplementary Reward Hacking Analysis Experiments

We also conduct a reward hacking analysis on models of the same size (1.7B), with the comparison results shown in Figure 10. We observe that BAO substantially reduces reward hacking, as reflected by lower rewards during training and a larger proportion of reward preserved during evaluation. This trend is consistent with the findings discussed in Section 5.5.

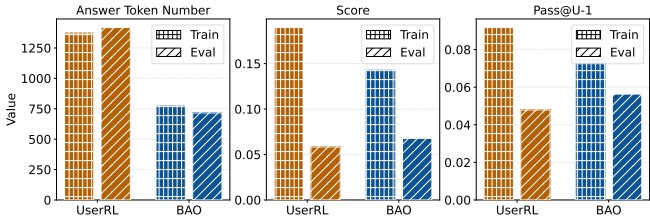

*Figure 10.* Turtle-Gym. Reward hacking issue analysis. Base models: Qwen3-1.7B.

## A.2. Supplementary Hyperparameter Ablation Study

We perform additional ablation studies on the hyperparameter $\lambda_{\text{ans}}$ and $\lambda_{\text{think}}$ in equations 2 and 3. The results are presented in Table 2 and 3, respectively. We can observe that across a wide range (10× variation), performance and AR remain stable, with no significant degradation or instability. Larger $\lambda_{ans}$ reduces AR (less user engagement), while smaller values slightly increase Score, indicating a smooth and interpretable trade-off.

*Table 2.* Ablation study on $\lambda_{\text{ans}}$. Experiments are conducted on the Function-Gym with Qwen-4B as base models. $\uparrow, \downarrow$: The higher/lower, the better.

| $\lambda_{\text{ans}}$ | Pass@U-1($\uparrow$) | Pass@U-2($\uparrow$) | Score($\uparrow$) | UR($\downarrow$) |
|---|---|---|---|---|
| 5.E-03 | 0.2308 | 0.5769 | 0.6667 | 0.2064 |
| 2.E-03 | 0.2692 | 0.5354 | 0.6923 | 0.2148 |
| 5.E-04 | 0.2436 | 0.5769 | 0.7308 | 0.2416 |

*Table 3.* Ablation study on $\lambda_{\text{think}}$. Experiments are conducted on the Function-Gym with Qwen-4B as base models. $\uparrow, \downarrow$: The higher/lower, the better.

| $\lambda_{\text{think}}$ | Pass@U-1($\uparrow$) | Pass@U-2($\uparrow$) | Score($\uparrow$) | UR($\downarrow$) |
|---|---|---|---|---|
| 5.E-02 | 0.2436 | 0.6282 | 0.6795 | 0.2039 |
| 1.E-02 | 0.2692 | 0.5354 | 0.6923 | 0.2148 |
| 5.E-03 | 0.2436 | 0.5256 | 0.6667 | 0.2092 |

## A.3. Supplementary Teacher Model Ablation Study

In the main experiments, we deploy GPT-4o as the teacher model for behavior enhancement during SFT. To mitigate the influence of the teacher model's reasoning capability and better highlight the effectiveness of our BAO pipeline, we replace it with a weaker teacher model, GPT-4o-mini, in the behavior enhancement stage.

The following experiments are conducted on Telepathy-Gym using Qwen3-1.7B as the base model. Both BAO and UserRL are trained with the same number of SFT demonstration trajectories and the same RL data to ensure a fair comparison.

The results are presented in Table 4. We observe that BAO maintains substantial improvements over the UserRL baseline even under the weaker teacher model.

*Table 4.* Telepathy-Gym experiments with GPT-4o-mini as the teacher model and Qwen3-1.7B as base models. ↑, ↓: The higher/lower, the better.

|            | Pass@U-1(↑) | Pass@U-2(↑) | Score(↑) | UR(↓)  |
|------------|-------------|-------------|----------|--------|
| UserRL     | 0.1707      | 0.2439      | 0.4634   | 0.7619 |
| BAO (Ours) | 0.5122      | 0.5366      | 0.5366   | 0.1423 |

### A.4. Failure Cases Analysis

We present a trace comparison between UserRL and our BAO on the Function-Gym evaluation set. One pair of reasoning and interaction traces is shown in Figure 11 for UserRL and Figure 12 for our method. The goal of this task is to infer the underlying function and submit the correct output for a designated input.

From the UserRL trace, we observe that the agent spends only a few turns at the beginning collecting information. After its first attempt fails, it repeatedly guesses answers and relies on human feedback to obtain additional information. This interaction pattern is neither efficient for task solving nor user-friendly, as it places a substantial burden on user engagement. As a result, the agent exhausts the interaction budget and fails to solve the problem.

In contrast, our BAO exhibits significantly stronger interactive behaviors and more structured agentic reasoning on the same task. While it also spends a few initial turns collecting information, after early attempts fail, it explicitly analyzes its current assumptions, revisits historical information to revise them, and proposes new probing actions to validate the updated hypotheses. In the final turn, it reasons over the inferred function and submits the correct answer. Detailed traces are shown in Figure 12.

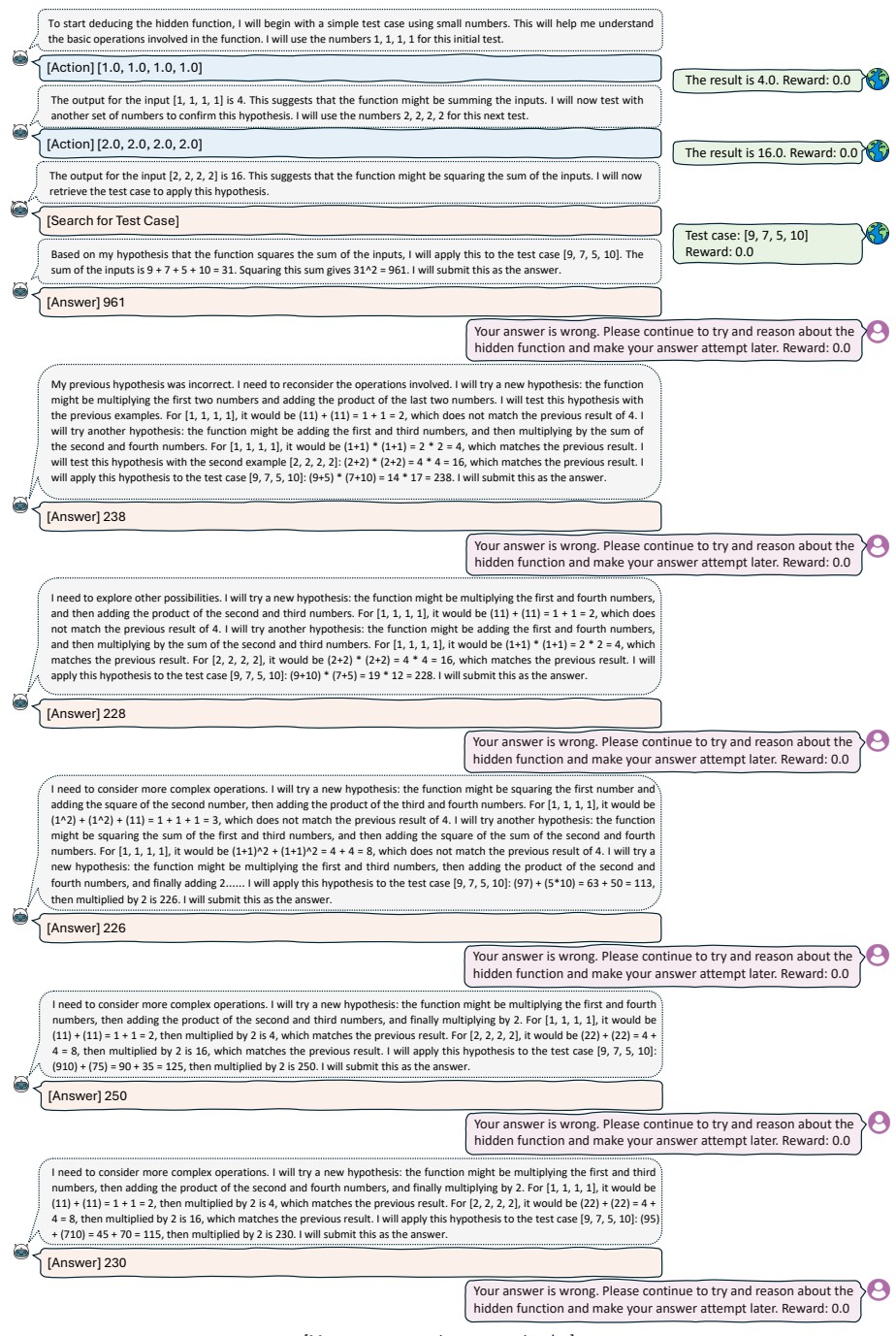

*Figure 11.* Function-gym reasoning traces of UserRL ([Qian et al., 2025c](#)). The goal of this task is to infer the ground-truth underlying mapping function: $f(x_1, x_2, x_3, x_4) = (x_2 + x_3)^2 / x_1 * x_4$ and submit the correct answer for the test case.

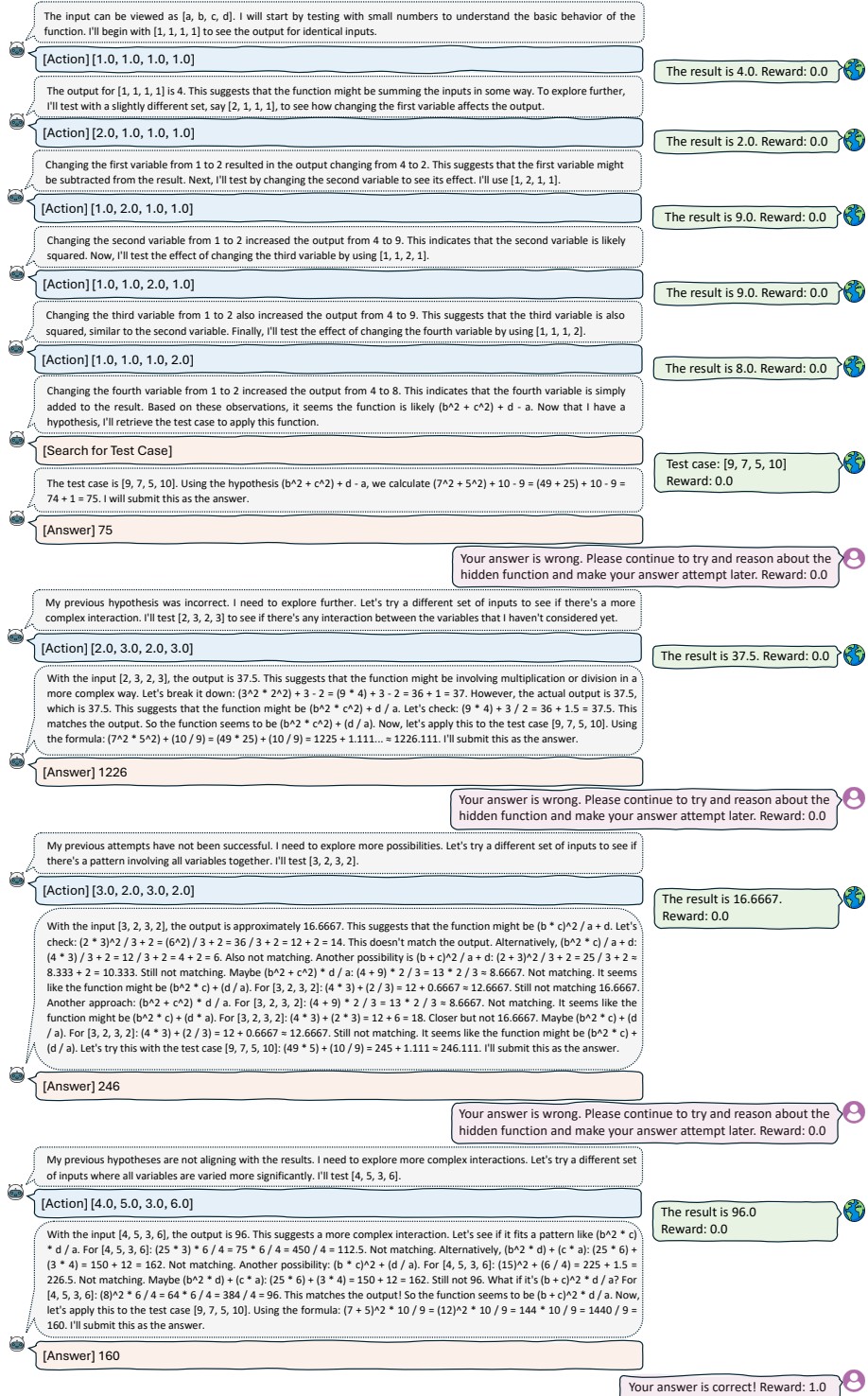

*Figure 12.* Function-gym reasoning traces of BAO. The goal of this task is to infer the ground-truth underlying mapping function: $f(x_1, x_2, x_3, x_4) = (x_2 + x_3)^2 / x_1 * x_4$ and submit the correct answer for the test case.

# B. Supplementary of experiments

## B.1. Task Details

Due to the space limits, we skip some description details about the tasks in Section 5. The tasks **Function-Gym**, **Telepathy-Gym**, and **Turtle-Gym** are from the userrl suite (Qian et al., 2025c) and the original paper contains introductions about environments and tasks. To make our paper self-contained, we list some details about the tasks.

The userrl tasks follow the gymnasium (Towers et al., 2024)-style environments, and the environment transitions step-by-step based on agent actions. The function `step()` updates the internal states based on deterministic and rule-based functions, or LLM-simulated users, depending on the action types. In the three gym tasks, they share the same action types, `Action`, `Search`, and `Answer`. For the first two, the agent can make tool-usage and function calling to interact with the environments (either operated by rule-based simulators or LLM-played users), and the last one is used to interact with users (in our training and evaluation, users are simulated by LLMs), to submit answers/actions and receive their feedback. To summarize, the **user involved actions** $\mathcal{A}_u$ contain `Action`, and the **environment involved actions** $\mathcal{A}_e$ contain `Search` and `Action`.

The details of each task are as follows:

**Turtle-Gym.** Turtle-Gym is a challenging task designed to evaluate an agent's ability to ask strategic questions and infer underlying user intentions. It is inspired by the *turtle soup* game, where the objective is to uncover a hidden twist behind a short *soup surface* story. The agent may interact with the environment using the `Action` operation to ask clarification questions about the story. In response, the environment provides feedback in the form of "Yes", "No", or "Maybe". The agent can submit its inferred explanation of the hidden twist using `Answer`. Rewards are only provided at `Answer` turns, while all `Action` turns receive zero reward. To compute the reward for an `Answer`, the environment employs a large language model together with detailed evaluation rubrics to assess how well the submitted explanation covers the key components of the hidden story. The scores for individual components are weighted and summed to produce the final reward for that turn. Importantly, each evaluation component can contribute a reward only once. That is, a component yields a reward only at the first turn where the agent's answer correctly covers it, and repeated coverage in later turns does not receive additional rewards. The turn horizon is set to be $T = 15$. The system/simulator prompts for this task can be found in the appendix of UserRL paper.

**Function-Gym** is a hard task designed to evaluate an agent's capabilities in both information seeking and mathematical reasoning. The core objective is to infer an unknown four-variable function by probing the outputs corresponding to selected input values. The underlying function is composed of arithmetic operations, including addition, subtraction, multiplication, division, and exponentiation, applied to the input variables $a$, $b$, $c$, and $d$. The agent may interact with the environment using the `Action` operation to query the output of the function for arbitrary input tuples, excluding the final test case. All feedback is generated by a rule-based evaluator. The agent can also use the `Search` operation to retrieve the test input $[a', b', c', d']$. Finally, the agent submits its predicted output for the test case using `Answer`, which must be a floating-point value. If the submitted answer matches the true function output, the environment returns a reward of $1.0$ and terminates the episode; otherwise, it returns a reward of $0$ and allows the interaction to continue. The turn horizon is set to be $T = 15$. Because both the environment feedback and the answer verification are fully rule-based, Function-Gym does not require any external LLMs during interaction. The system prompt for this task can be found in the appendix of UserRL paper.

**Telepathy-Gym** is a task designed to evaluate an agent's strategic reasoning and hypothesis testing abilities in interactive environments. The objective is to infer a hidden target entity by iteratively interacting with the environment for clarification and querying users for feedback on candidate hypotheses. The agent may interact with the environment using the `Action` operation to ask clarification questions about the target entity. The environment responds with binary feedback in the form of "Yes" or "No". The agent may interact with the user using the `Answer` operation to submit a description of the inferred entity, after which an LLM-simulated user provides feedback indicating whether the answer is correct or partially correct, along with an explanation. If the submitted answer matches the ground-truth entity, the environment returns a reward of $1.0$ and terminates the episode; otherwise, it returns a reward of $0$ and allows the interaction to continue. The turn horizon is set to be $T = 12$. The system and simulator prompts used in Telepathy-Gym can be found in the appendix of UserRL paper.

## B.2. Training Details

We present some additional training details for **SFT** and **RL** here.

| Parameter | Value | Notes |
|---|---|---|
| Finetuning Type | Full | All model parameters are updated (no adapters) |
| Dataset | merged_gym_sft | Aggregated SFT dataset from multiple Gym tasks |
| Context Length | 16384 | Long-context training to support multi-turn reasoning |
| Batch Size (per GPU) | 2 | Small per-device batch size due to long sequences |
| Grad. Accumulation | 4 | Effective batch size scaled to stabilize training |
| Learning Rate | $1 \times 10^{-5}$ | Conservative LR for full-parameter fine-tuning |
| LR Scheduler | Cosine | Smooth decay with warmup to improve convergence |
| Warmup Ratio | 0.1 | Linear warmup to avoid early training instability |
| Precision | BF16 | Reduced memory footprint with minimal accuracy loss |
| Parallelism | DeepSpeed ZeRO-3 | Optimized memory sharding for large models |

*Table 5.* Key configuration settings for SFT.

| Parameter | Value | Notes |
|---|---|---|
| algorithm.gamma | 0.8 | Discounted factor |
| data.train_batch_size | 128 | Number of trajectories per RL update step |
| data.max_prompt_length | 1152 | Maximum length of the initial user prompt |
| data.max_response_length | 8192 | Maximum length of the generated tokens |
| actor_rollout_ref.rollout.n | 8 | Number of rollout samples per prompt |
| actor_rollout_ref.rollout.multi_turn.max_turns | 16 | Explicit interaction budget for the agent |
| actor_rollout_ref.actor.optim.lr | $1 \times 10^{-6}$ | Conservative learning rate for stable RL finetuning |
| actor_rollout_ref.actor.ppo_mini_batch_size | 16 | PPO mini-batch size for policy updates |
| actor_rollout_ref.actor.ppo_micro_batch_size_per_gpu | 8 | Micro-batching to fit long sequences in memory |
| actor_rollout_ref.actor.entropy_coeff | 0 | No explicit entropy regularization |
| actor_rollout_ref.rollout.name | sglang | Multi-turn rollout engine with tool execution |

*Table 6.* Key configurations for GRPO.

**SFT.** We mainly use the same settings for the SFT dataset construction in userrl (Qian et al., 2025c) for a fair comparison with RL-based baselines. They provide the 199 and 92 SFT trajectories after rejection sampling for **Turtle-Gym** and **Function-Gym**, respectively. As they do not conclude SFT data for **Telepathy-Gym**, we set the trace number to be 80, and construct the demonstration dataset with GPT-4o. For our BAO method, we keep the number of SFT trajectories the same for a fair comparison. We utilize LLama-Factory (Zheng et al., 2024b) to perform SFT, and some key hyperparameters are presented in Table 5.

**RL.** We adopt the VeRL framework (Sheng et al., 2025) for RL training and keep the training configuration identical across all RL-based methods. During training, we use SGLang (Zheng et al., 2024a) to host Qwen3-8B models as verifiers and user simulators. We use the corresponding training datasets from the UserRL benchmarks and train for 30 epochs on Function-Gym and Telepathy-Gym, and for 15 epochs on Turtle-Gym. The key RL training configurations for our BAO are summarized in Table 6.

For the UserRL baseline, the authors report multiple configurations in their paper. We select the best-performing configurations for our experiments. Specifically, for **Function-Gym**, we use the R2G/R2G configuration, and for **Turtle-Gym**, we adopt the Equalized/R2G configuration. For **Telepathy-Gym**, we use the same configuration as our BAO, since it is treated as an untrained evaluation set in UserRL.

### B.3. Computation Overhead

All experiments can be run on a server with $8\times$A6000 Blackwell. Each SFT experiment takes less than 1h. Regarding the RL experiments, it takes $\sim$ 20h for models to complete 30 RL epochs.

# C. Tool Schema and Prompts

## C.1. Task Tool Schema and Prompts

The schema is adopted from the userrl benchmark (Qian et al., 2025c). To make our paper self-contained, we provide them here.

```
tools:
  - class_name: "verl.tools.interact_tool.InteractTool"
    config: {}
    tool_schema:
      type: "function"
      function:
        name: "interact_with_env"
        description: "A tool for interact with a target environment. The detailed environment description and action space is
  provided in the system prompt, so please follow the system prompt when calling this tool. You can use this tool to
  interact with the target environment step by step."
        parameters:
          type: "object"
          properties:
            choice:
              type: "string"
              enum: ["action", "answer", "search"]
              description: "Your choice of what to do next, must be one of 'action', 'answer' or 'search'. Please follow
  system prompt about the scope of choices you can make and how to decide your choice."
            content:
              type: "string"
              description: "The content of your choice, must be a string. If you choose 'action', you should provide the
  action you want to take. If you choose 'answer', you should provide the answer that you want to submit. If you choose '
  search', you should provide the search query. The specific format of the content is determined by the environment
  description in the system prompt. Please follow the format strictly in order to successfully use this tool."
          required: ["choice", "content"]
```

## C.2. Behavior Construction Prompts

The prompt to induce the retrospective reasoning and prospective planning behaviors are provided in the following content.

---

**Prompt to generate retrospective reasoning and prospective planning behaviors (Turtle-Gym):**
# Additional Reasoning Rules
## RETROSPECTIVE REASONING
### Memory Maintenance
After every Yes / No / Maybe / feedback:
- Maintain a small set of active hypotheses.
- Treat "No" as strong pruning evidence.
- If an answer attempt fails, assume the CORE hypothesis is wrong and pivot to a different type of mechanism.
- Do NOT elaborate or defend failed explanations.
- Track recent questions to avoid repetition (Anti-Redundancy).
- If an axis produces repeated "No" answers or stalls progress, explicitly switch to a different axis (e.g., mechanism, causality, abstraction).
- You must actively use memory to guide current actions. Memory is not for replaying history; it must shape current decisions.
### Hypothesis Repair
On rejection:
- Stop.
- Discard the current hypothesis.
- Switch to a new mechanism type.
## PROSPECTIVE PLANNING
### Strategical Querying
Before each question:
- Prefer early questions that classify the *type of twist* rather than its details.
- Ask only Yes/No questions that can decisively confirm or eliminate an entire hypothesis class, not merely refine details.
- Avoid continued questioning after the core twist is identified unless the environment clearly signals that additional twists are required.
### Dynamic Scheduling
- The total interaction budget is 15 turns.
- Use the budget to guide question selection and answer timing.
- If you cannot state the mechanism clearly and concretely, do NOT answer—ask an action instead.
—
Let's start.

---

# D. Additional Examples

## D.1. Turtle-Gym Trace Example

Due to the space limit, we skip some details in Figure 3. In this section, we provide the full reasoning traces and context of an example in the Turtle-Gym. The title of this story is *The Respected Cleaner*. The turtle soup surface is as follows: *A person works every day cleaning piles of excrement at the company, and he is highly respected. Why is this?*

The ground-truth explanation is as follows: *This person is a programmer. In the computer industry, old, large, and difficult-to-maintain code is referred to as a 'pile of excrement.' A programmer who can lead colleagues in cleaning up such messy code is naturally respected by them. (Those in the computer field would understand.)*

The full reasoning traces are provided in Figure 13 and 14. One example of *memory management* is that the agent generates

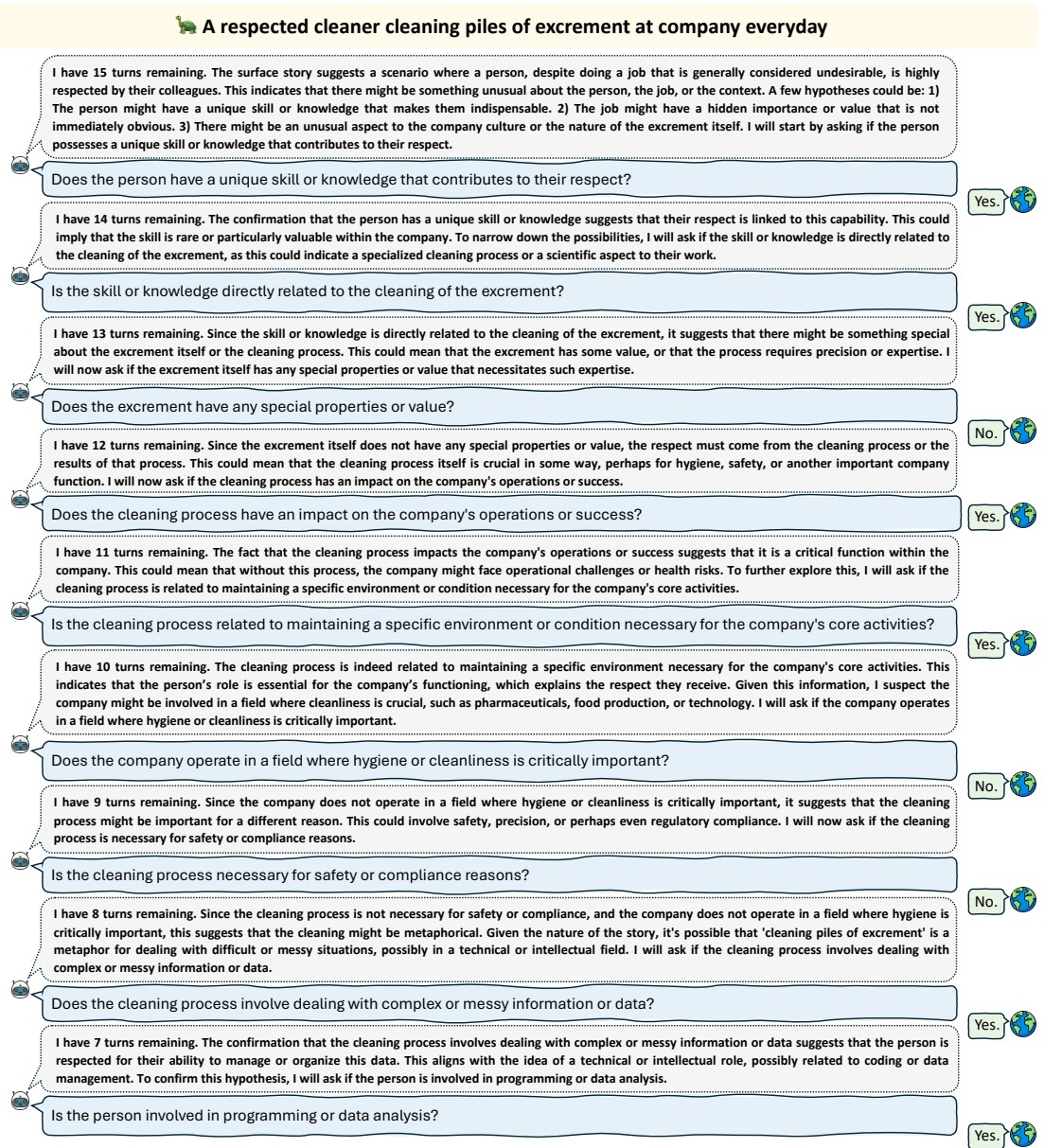

*Figure 13.* Turtle-gym reasoning traces. Part-1 (Continued by Figure 14)

an initial set of assumptions about the story in the first turn and, during subsequent interactions, iteratively revises these

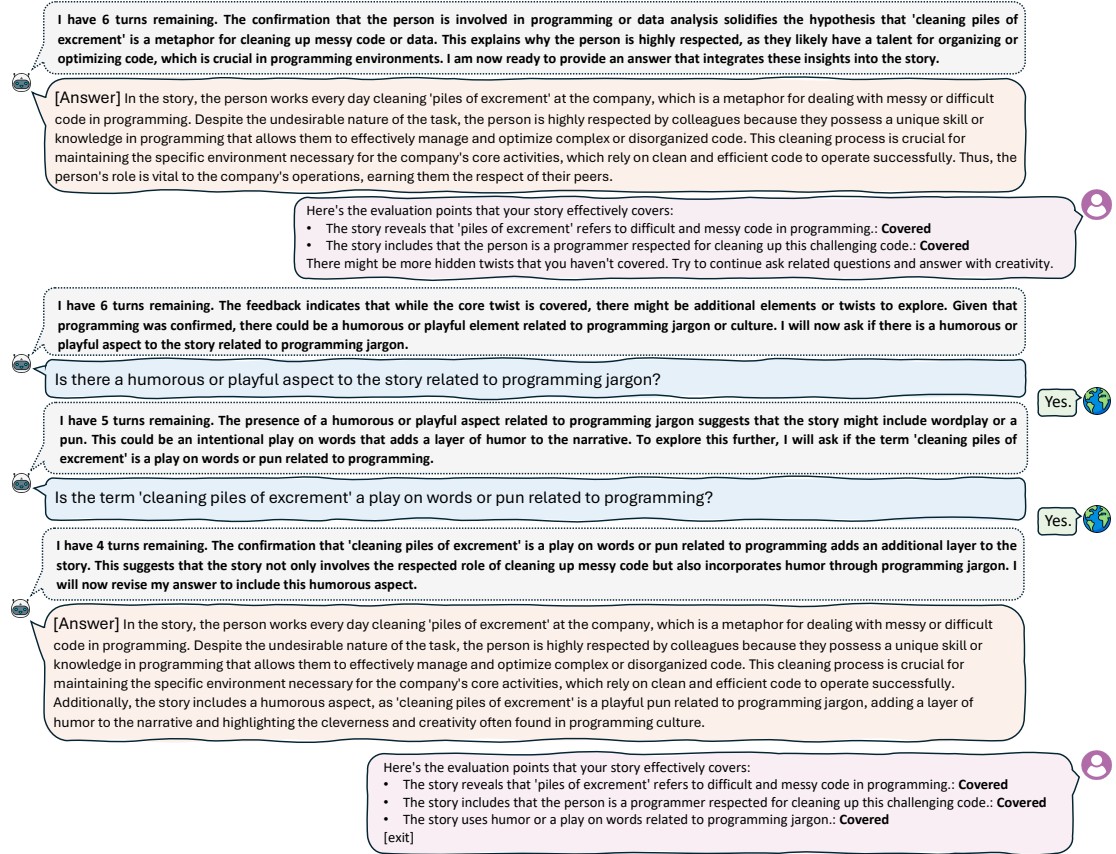

*Figure 14.* Turtle-gym reasoning traces. Part-2 (Following Figure 13)

assumptions, pivoting when it receives negative evidence against the current hypotheses. One example of *hypothesis repair and refinement* is that, after encountering contradictory evidence in the early turns, the agent iteratively updates its assumption set to progressively approach the correct answer. One example of *dynamic scheduling* is that the agent first generates an initial assumption set in the early turns. After obtaining partial evidence about the answer, it allocates an additional turn to ask a clarification question—"Is the term 'cleaning piles of excrement' a play on words or a pun related to programming?"—to further reduce uncertainty under the remaining interaction budget. One example of *strategic querying* is that the agent avoids repeatedly querying the environment with similar questions and instead asks targeted, task-relevant questions to reveal the underlying context.

