# OpenReview forum: "Pushing Forward Pareto Frontiers of Proactive Agents with Behavioral Agentic Optimization"
_ICML.cc/2026/Conference — ICML 2026 regular_

### Official Review · Reviewer_CXnk · 2026-03-03

**Soundness:** 3
**Presentation:** 3
**Significance:** 2
**Originality:** 3
**Overall Recommendation:** 4
**Confidence:** 3

**Summary:**

This paper addresses the inherent trade-off between task performance and user engagement (measured by answer rate) in proactive LLM agents, and formulates this challenge through the lens of Pareto frontier optimization. The authors demonstrate that simply adjusting scalar reward weights for the two objectives is insufficient to meaningfully improve the trade-off.

To overcome this limitation, they propose a two-stage framework called Behavioral Agentic Optimization (BAO). In the SFT stage, they introduce behavior enhancement mechanisms, explicitly incorporating Retrospective Reasoning and Prospective Planning to improve multi-turn reasoning structure. In the RL stage, they apply behavioral regularization to encourage effective information-seeking while penalizing excessive internal overthinking.

Experiments on three tasks from the UserRL benchmark, along with comprehensive ablation studies, demonstrate that this two-stage approach improves the trade-off between task performance and user engagement, effectively pushing the Pareto frontier forward.

**Compliance With Llm Reviewing Policy:**

Affirmed.

**Key Questions For Authors:**

See the weaknesses above.

**Limitations:**

yes

**Strengths And Weaknesses:**

Strengths
1. The paper addresses an important and timely problem: the trade-off between task performance and user engagement in proactive LLM agents. Framing this as a Pareto frontier problem is conceptually clean and helps explain why simple reward reweighting is insufficient.

2. The paper is easy to follow. The proposed behavioral methods are intuitively grounded in interactive reasoning, and the two-stage pipeline is relatively straightforward to replicate.

3. The evaluation includes strong ablations and reward hacking analysis, supporting the effectiveness of each component.

Weaknesses

1. While the Pareto framing is interesting, it is not fully clear why the proposed behavioral methods are uniquely suited to extending the frontier. Other approaches, such as personalization or better user modeling as cited, might also improve the Pareto frontier.  Adding such baselines for comparison or further discussion would strengthen the paper.

2. Experiments are limited to three UserRL tasks, Although these settings are diverse and well-designed, they are still relatively small-scale and somewhat synthetic. Demonstrating effectiveness in more practical domains (e.g., real-world coding or SWE scenarios[1] ) would improve the paper’s impact and generalizability.

In summary, this is a clear and engineering-wise informative paper. I would be willing to increase my score if the above concerns are adequately addressed.

[1] Training Proactive and Personalized LLM Agents, https://arxiv.org/pdf/2511.02208

######update#####

I have increased the score.

---

> ### Author Rebuttal · Authors · 2026-03-30
>
> We thank the reviewer for the useful comments and provide rebuttals as follows.
>
> - W1: Comparison to alternative approaches
>
>     We thank the reviewer for this important point. In our experiments, we compared our method with a **strong user-centric RL baseline, UserRL** [1], which models user preferences and updates its policy based on both task performance and user modeling accuracy. Compared to this baseline, our method emphasizes the importance of multi-turn behaviors, including retrospective reasoning and prospective planning, in proactive agents, and enforces these behaviors in both the SFT and RL stages. These behaviors enable more efficient interaction with users, improve information analysis and collection, and lead to better task performance while reducing unnecessary user engagement.
>
>     BAO is **orthogonal to user modeling or personalization** methods. The behaviors introduced in BAO improve how the agent interacts with users and the environment, regardless of the specific user modeling methods. In contrast, personalization works such as PersonaMem [2] rely on dynamic user profiling and personalized response generation, where the model **leverages historical interaction context to infer user preferences** and generate appropriate responses. We believe these directions are complementary and that combining them is a promising direction for future work.
>
>     To further examine whether simpler approaches can also improve the Pareto frontier, we introduce **an additional baseline** inspired by **CollabLLM** [3], which penalizes user interaction via intrinsic reward. To ensure a fair comparison, we upgrade DPO with GRPO and denote this variant as CollabLLM-GRPO.
>
>     Results on Function-Gym (4B base models):
>
>     ||Score($\uparrow$)|AR($\downarrow$)|
>     |-|-|-|
>     |UserRL|0.5256|0.3758|
>     |CollabLLM-GRPO|0.4872|0.2917|
>     |BAO|0.6923|0.2148|
>
>     We observe that simply penalizing user interaction reduces AR (Answer Rate, which represents the user engagement effort)  but also degrades task performance, leading to an over-conservative policy that behaves more passively. This confirms that naive reward penalties cannot effectively improve the trade-off.
>
>     In contrast, BAO improves the score and reduces AR simultaneously, indicating a solid improvement of the Pareto frontier over task performance and user engagement. This suggests that the key factor is not merely penalizing user interaction, but enabling the agent to strategically collect and utilize information via structured behaviors.
>
>     We have revised our paper to include the discussions above.
>
> - W2: Additional evaluation on practical tasks
>
>     We further evaluate BAO on UserBench [4], a realistic benchmark that requires agents to proactively clarify user intent and make grounded decisions using tools in travel scheduling tasks. The tasks include booking flights, hotels, restaurants, and other common travel-related activities.
>
>     |Model|Gemini-2.5-Pro|GPT-4o|UserRL-4B|BAO-4B|
>     |-|-|-|-|-|
>     |Score($\uparrow$)|0.3468|0.3643|0.5086|0.6145|
>
>     We maintain strictly controlled training conditions (same RL data and training epochs as UserRL). BAO consistently improves over the RL baseline and remains competitive with strong closed-source models.
>
>     These results demonstrate that BAO generalizes to more realistic user-centric tasks.
>
>     Reference:
>
>     [1] Cheng Qian, et al. "Userrl: Training interactive user-centric agent via reinforcement learning." arXiv preprint arXiv:2509.19736 (2025).
>
>     [2] Bowen Jiang, et al. "Know me, respond to me: Benchmarking LLMs for dynamic user profiling and personalized responses at scale." COLM 2025.
>
>     [3] Shirley Wu, et al. "Collabllm: From passive responders to active collaborators." ICML 2025
>
>     [4] Cheng Qian, et al. "Userbench: An interactive gym environment for user-centric agents." arXiv preprint arXiv:2507.22034 (2025).

---

> > ### Author Rebuttal · Reviewer_CXnk · 2026-03-31
> >
> > Authors' response solved most of my questions. I have increased my score.

---

> > > ### Author Response · Authors · 2026-04-06
> > >
> > > We thank the reviewer for the time and effort throughout the review process, and we sincerely appreciate the helpful feedback and comments on our paper. We will incorporate the points from the rebuttal into the revised manuscript.

---

### Official Review · Reviewer_ptrd · 2026-03-04

**Soundness:** 2
**Presentation:** 3
**Significance:** 3
**Originality:** 2
**Overall Recommendation:** 4
**Confidence:** 3

**Summary:**

This paper investigates the trade-off between task performance and user burden in multi-turn interactions for proactive agents. The authors propose a Behavioral Agentic Optimization framework, which first uses a teacher model to generate trajectories with retrospective reasoning and prospective planning for supervised fine-tuning. Then, behavioral regularization is incorporated during the reinforcement learning phase to suppress excessive user validation and ineffective thinking, and the GRPO update strategy is used for optimization. Experiments simultaneously evaluate success rate and user engagement in multiple interaction environments, demonstrating that this method can improve or maintain task performance while reducing user involvement.

**Compliance With Llm Reviewing Policy:**

Affirmed.

**Final Justification:**

The rebuttal substantially addressed my main concerns: the newly provided hyperparameter sensitivity analysis demonstrates stable performance, and the cross-task evaluation on UserBench reinforces the method's generalizability. I am therefore upgrading my score to 4

**Key Questions For Authors:**

1.The authors' method incorporates multiple manual penalty terms into the scalar reward to guide behavior. How did the authors assess the robustness of this strategy? For example, will significant instability or performance fluctuations occur under different tasks or different random seeds?

2.The paper lacks analysis of key hyperparameters such as $\lambda_{ans}$ and $\lambda_{think}$. Could the authors provide a basic sensitivity experiment, such as scanning these parameters within a reasonable range and reporting the trends in performance and answer rate, thus providing a relatively stable usable range and improving reproducibility?

3.Please supplement the key arguments in the introduction with relevant literature (i.e., paragraph 3).

If the authors address all the questions, a higher score may be considered.

**Limitations:**

see weakness

**Strengths And Weaknesses:**

1.Strength

a. This paper is well-written and clearly presented.

b. The experimental analysis is quite comprehensive. In addition to the main table results and module ablation, the authors also supplemented the analysis with explorations of diversity and reward hacking to explain why BAO behavior is better.

c. From the outset, the paper presents "performance" and "user engagement cost" as parallel objectives, consistently using metrics such as Pass@A-k, Score, and Answer Rate to characterize both in the experiments. Finally, a Pareto frontier is used to unify the trade-off changes. This narrative is self-contained and makes sense.

2.Weakness

a. The authors point out that a single-weighted scalar objective is insufficient to improve the trade-off between performance and user cost, but their method essentially still relies on adding an additional manual penalty term to the scalar reward. This approach of forcing the model to exhibit a certain behavior by finely adjusting the manual reward term often leads to a fragile strategy; I suspect it may be particularly sensitive to hyperparameters.

b. Based on the above issues, I tried to find relevant hyperparameter experiments in the article, such as $\lambda_{ans}$, $\lambda_{think}$, etc., but unfortunately I couldn't find any. I think the authors should at least analyze these parameters to find a reasonable range for parameter selection, which would also enhance the reproducibility of the conclusions.

c. The introduction states, "...However, this comes at a cost: repeated or redundant requests for human engagement can erode user confidence in the agent’s competence...." This is one of the core motivations of this paper, but the introduction does not provide relevant references to support it, which I think is still very necessary.

---

> ### Author Rebuttal · Authors · 2026-03-30
>
> We thank the reviewer for the useful comments and provide rebuttals as follows.
>
> - W-a, W-b, Q2: Hyperparameter sensitivity and robustness
>
>     (1) Sensitivity analysis.
>
>     - We conduct additional ablations on Function-Gym (4B base models). In our main experiments, we use $\lambda_{ans}=2e-3$ and $\lambda_{think}=1e-3$.
>
>         |$\lambda_{ans}$|Pass@A-1($\uparrow$)|Pass@A-2($\uparrow$)|Score($\uparrow$)|AR($\downarrow$)|
>         |-|-|-|-|-|
>         |5e-3|0.2308|0.5769|0.6667|0.2064|
>         |2e-3|0.2692|0.5354|0.6923|0.2148|
>         |5e-4|0.2436|0.5769|0.7308|0.2341|
>
>         |$\lambda_{think}$|Pass@A-1($\uparrow$)|Pass@A-2($\uparrow$)|Score($\uparrow$)|AR($\downarrow$)|
>         |-|-|-|-|-|
>         |5e-3|0.2436|0.6282|0.6795|0.2039|
>         |1e-3|0.2692|0.5354|0.6923|0.2148|
>         |5e-4|0.2436|0.5256|0.6667|0.2092|
>
>         Across a wide range (10× variation), performance and AR remain stable, with **no significant degradation or instability**. Larger $\lambda_{ans}$ reduces AR (less user engagement), while smaller values slightly increase Score, indicating a smooth and interpretable trade-off.
>
>     (2) Robustness across seeds and tasks.
>
>     - We report results averaged over 3 seeds (Fig. 5). For example:
>
>         ||Score($\uparrow$)|AR($\downarrow$)|
>         |-|-|-|
>         |UserRL-1.7B|0.2778 $\pm$ 0.0121|0.2845 $\pm$ 0.0270|
>         |BAO-1.7B|0.3633 $\pm$ 0.0160|0.2148 $\pm$ 0.0139|
>     BAO consistently improves both metrics with low variance.
>
>     To evaluate the robustness on other tasks, we also perform evaluations on **UserBench** [1], which is a realistic benchmark that requires agents to proactively clarify user intent and make grounded decisions using tools in travel scheduling tasks. The tasks include booking flights, hotels, restaurants, and other common travel-related activities.
>
>     |Model|Gemini-2.5-Pro|GPT-4o|UserRL-4B|BAO-4B|
>     |-|-|-|-|-|
>     |Score|0.3468|0.3643|0.5086|0.6145|
>
>     We maintain strictly controlled training conditions (same RL data and epochs as UserRL). BAO consistently improves over the RL baseline and is competitive with strong closed-source models. This result suggests that **BAO’s improvements are generic under realistic, user-centric tasks** beyond the three benchmarks reported in the paper.
>
>     (3) Beyond scalar reward tuning.
>
>     While BAO uses reward shaping, it is not equivalent to simple weight tuning. As shown in Fig. 2, adjusting the scalar weight alone fails to improve the Pareto trade-off. In contrast, BAO introduces behavioral priors (retrospective reasoning and planning) via SFT and structural regularization during RL, which changes exploration patterns (Sec. 5.4) and leads to consistent Pareto improvements.
>     Thus, **the gains arise from behavior-level optimization, not only reward scaling.**
>
>
> - W-c: Missing references in Introduction
>
>     We agree and will add supporting references. Prior work has highlighted that requiring frequent user interaction can reduce interaction efficiency and increase user burden. For example, proactive agent literature notes that reactive systems require users to continuously provide explicit inputs, which can hinder the flow of interaction [2]. Works such as CollabLLM [3] emphasize the importance of optimizing interaction efficiency in multi-turn human-AI collaboration.
>
>     Reference:
>
>     [1] Cheng Qian, et al. "Userbench: An interactive gym environment for user-centric agents." arXiv preprint arXiv:2507.22034 (2025).
>
>     [2] Yaxi Lu, et al. "Proactive agent: Shifting llm agents from reactive responses to active assistance." arXiv preprint arXiv:2410.12361 (2024).
>
>     [3] Shirley Wu, et al. "Collabllm: From passive responders to active collaborators." ICML 2025

---

### Official Review · Reviewer_S5tx · 2026-03-09

**Soundness:** 3
**Presentation:** 3
**Significance:** 3
**Originality:** 3
**Overall Recommendation:** 5
**Confidence:** 3

**Summary:**

This paper is about training proactive, multi-turn LLM agents that can ask questions and use tools, but do so without constantly pulling the user into the loop. The authors frame this as a Pareto trade-off between task success (reward) and user engagement cost (how often the agent uses “user-involved” actions).
They propose BAO, which (1) injects two inter-turn behaviors—retrospection (track/adjust hypotheses, manage memory) and prospection (plan steps, budget queries)—via SFT on teacher-generated traces, and then (2) runs RL with simple regularizers that discourage consecutive user-interaction actions and “over-thinking” failures.
They evaluate on three UserRL tasks and report better success–engagement trade-offs plus some analysis on exploration and reward hacking.

**Compliance With Llm Reviewing Policy:**

Affirmed.

**Final Justification:**

Authors improved soundness of the paper and answered my concerns

**Key Questions For Authors:**

1. For the “no information gain” penalty: is it intentionally a consecutive user-action heuristic, or is there any additional signal used to detect “information gain”? Please clarify to align the text with the implementation.
2. What specific failure/termination conditions trigger the “over-thinking” penalty, and how do you distinguish them from normal early termination or environment-driven endings?

**Limitations:**

Not fully. The limitations section would be stronger with more concrete discussion of:

- how results may differ with real users
- how the approach behaves with different tool ecosystems or longer/harder tasks

**Strengths And Weaknesses:**

1. Soundness

Strengths:
- The paper presents a reasonable and practical framing of “agent quality” that includes both success and user burden, and uses Pareto trade-offs to reflect different deployment preferences.
- The two-stage approach (SFT for behaviors + RL to refine interaction strategy) is coherent, and the paper includes ablations indicating both parts contribute.
- The paper adds useful analysis beyond headline scores (e.g., exploration and reward-hacking observations), which helps interpret results.

Weaknesses:
- There appears to be some inconsistency in how action types are categorized (which actions count as user-involved vs. environment/tool actions). Since the engagement objective and penalties depend on this mapping, it affects interpretability and reproducibility.
- Information-seeking Regularizer is motivated as discouraging repeated user interaction “without information gain,” but the described trigger seems closer to a simpler heuristic (consecutive user-involved actions). Also not clear how information gain is measured. That’s still plausible, but the writeup could be more precise.
- Evaluation is limited to three benchmark environments (with simulated feedback in parts), so external validity to real-user settings and broader tool ecosystems remains uncertain.

2. Presentation

Strengths:
- Overall structure is easy to follow: problem framing → target behaviors → training recipe → results/ablations → analysis.
- Metrics are introduced and used consistently in the main experimental narrative.

Weaknesses:
- The action/engagement terminology could be tightened, especially to remove ambiguity around what counts as “user engagement” across tasks.
- Comparisons against “frontier/commercial” models would benefit from clearer caveats about differences in tuning and action-budget constraints.

3. Significance

Strengths:
- Reducing unnecessary user involvement while preserving performance is relevant for interactive assistants and tool-using agents.
- Emphasizing Pareto trade-offs is a practical reporting choice and may influence how agent evaluations are presented.

Weaknesses:
- The impact is currently demonstrated within a specific benchmark suite; stronger evidence of generalization would increase significance.

4. Originality

Strengths:
- The paper packages a clear set of inter-turn behaviors with a practical training pipeline and shows measurable effects on interaction efficiency.
- The Pareto-frontier emphasis is a helpful perspective for multi-objective agent post-training.

---

> ### Author Rebuttal · Authors · 2026-03-30
>
> We thank the reviewer for the useful comments and provide rebuttals as follows.
>
> - W1, W3: Limited evaluation / external validity
>
>     We understand reviewer's concern regarding generalization beyond the three benchmark environments.
>
>     To further evaluate external validity, we also perform evaluations on UserBench [1], which is a realistic benchmark that requires agents to proactively clarify user intent and make grounded decisions using tools in travel scheduling tasks. The tasks include booking flights, hotels, restaurants, and other common travel-related activities.
>
>     |Model|Gemini-2.5-Pro|GPT-4o|UserRL-4B|BAO-4B|
>     |-|-|-|-|-|
>     |Score|0.3468|0.3643|0.5086|0.6145|
>
>     We maintain strictly controlled training conditions (same RL data and epoches as UserRL). BAO consistently improves over the RL baseline and remains competitive with strong closed-source models. This result suggests that BAO’s improvements are generic under realistic, user-centric tasks beyond the three benchmarks reported in the paper.
>
>     Reference:
>
>     [1] Cheng Qian, et al. "Userbench: An interactive gym environment for user-centric agents." arXiv preprint arXiv:2507.22034 (2025).
>
> - W1: Action type definition and consistency
>
>     We agree that clearer terminology would improve readability.
>
>     In our formulation, we use a consistent definition across all tasks:
>     - User-involved actions ($A_u$): actions that require explicit user feedback (e.g., answer submission / verification)
>     - Environment-involved actions ($A_e$): actions interacting with tools or simulators without user burden (e.g., search)
>
>     This definition is consistently applied in:
>     - Problem formulation (Eq. 1)
>     - Behavior Regularization (Sec. 4.2)
>     - Evaluation Metrics (Answer Rate)
>
>     Detailed task-specific instantiations of these action types are also provided in Appendix B for completeness. We have revised the paper to restate this definition in the main text to reduce ambiguity.
>
> - W1, Q1: “No information gain” regularizer
>
>     In our implementation, the regularizer is intentionally a simple heuristic:
>     - we penalize consecutive user-involved actions, i.e., when $a_t, a_{t-1} \in A_u$
>
>     This serves as a proxy for discouraging redundant user queries without sufficient intermediate reasoning or information gathering.
>
>     We agree that the phrase “no information gain” was imprecise. We will revise it to reflect that: no explicit information gain signal is computed during training. Instead, we use this lightweight structural heuristic.
>
>     Separately, in evaluation (Sec. 5.4), we analyze information-seeking efficiency using diversity-based metrics (Self-BLEU and semantic embedding similarity), which empirically support improved behavior.
>
> - W2: Terminology
>
>     We introduce the definition of action types in Appendix B.1 and refer to it in Sec. 5.1. We have revised the draft to present this more explicitly and avoid potential ambiguity.
>
> - W2: Tuning budget in comparison with comercial models.
>
>     We tune the user-involved action budget |U| over a wider range and present the comparison on the FunctionGym task as follows:
>     |\|U\||1|2|3|4|5|
>     |-|-|-|-|-|-|
>     |Gemini2.5-Pro|0.2308|0.3333|0.3718|0.3718|0.3718|
>     |GPT-4o|0.1282|0.1282|0.1282|0.1410|0.1410|
>     |BAO-4B|0.2692|0.5354|0.6538|0.6795|0.6923|
>
>     We observe that, compared to GPT-4o and Gemini2.5-Pro, BAO interacts more effectively with users and achieves substantially higher task performance across all budgets.
>
> - Q2: Over-thinking penalty clarification
>
>     The over-thinking penalty is applied when:
>     - the trajectory terminates before reaching the interaction budget, and
>     - does not receive a successful terminal signal from the environment.
>
>    This typically corresponds to cases where the agent that spends excessive tokens on internal reasoning but fails to complete the task or produce a valid answer.
>
>     In contrast,
>     - Normal successful termination (correct answer),
>     - or environment-driven termination
>
>     do not incur this penalty.
>
>     We have revised the text to make this condition explicit.
>
>
> - Limitation discussion.
>
>     We agree and have expanded the discussion along three axes:
>
>     - Real users:
>     Current evaluations rely on simulated users. While BAO shows improved generalization under user distribution shift (Sec. 5.5), training and validation with real human user data remains an important future direction.
>     - Tool ecosystems:
>     Our experiments focus on relatively structured tool interactions. Extending BAO to more complex toolchains (e.g., web agents, terminal agents) may introduce additional challenges in credit assignment and exploration.
>     - Longer and harder tasks:
>     Current tasks have bounded horizons (T ≤ 15). Scaling to longer-horizon settings may require more sophisticated memory and planning mechanisms beyond the current design.

---

> > ### Author Rebuttal · Reviewer_S5tx · 2026-04-03
> >
> > Authors response answered my questions

---

> > > ### Author Response · Authors · 2026-04-06
> > >
> > > We thank the reviewer for the time and effort throughout the review process, and we sincerely appreciate the helpful feedback and comments on our paper. We will incorporate the points from the rebuttal into the revised manuscript.

---

### Official Review · Reviewer_oZ1v · 2026-03-10

**Soundness:** 2
**Presentation:** 2
**Significance:** 3
**Originality:** 2
**Overall Recommendation:** 4
**Confidence:** 3

**Summary:**

This paper studies the training of proactive LLM agents that actively interact with users and environments over multiple turns to complete tasks. The authors identify a trade-off between task performance, which benefits from more user interactions, and user engagement minimization, which prefers fewer interactions, and formulate this as a multi-objective optimization problem. They propose BAO (Behavioral Agentic Optimization), a two-stage framework consisting of: (1) behavior enhancement via SFT, where a teacher model (GPT-4o) generates training data enriched with retrospective reasoning, including memory management and hypothesis refinement, and prospective planning, including dynamic scheduling and strategic querying behaviors; and (2) behavior-regularized RL, which applies turn-level reward shaping to penalize inefficient information-seeking and over-thinking during GRPO optimization. Experiments on three tasks from the UserRL benchmark with Qwen3 1.7B and 4B models show that BAO achieves higher Pass@A-k rates with lower user engagement compared to the UserRL baseline, pushing the Pareto frontier forward. Additional analyses on exploration diversity and reward hacking mitigation are provided.

**Compliance With Llm Reviewing Policy:**

Affirmed.

**Final Justification:**

My main initial concern was the multi-objective optimization framing. While I still believe the "Pareto frontier" terminology does not precisely apply here — since BAO strictly dominates the baseline on both objectives rather than navigating a trade-off — I acknowledge that this is more of a framing/presentation issue than a fundamental flaw in the method itself. The practical contributions are sound and stand independently of the MOO narrative. Weighing the strong empirical results, thorough ablations, and responsive rebuttal, I vote for Weak Accept.

**Key Questions For Authors:**

1. The information-seeking regularization (Eq. 2) penalizes consecutive user actions regardless of whether information was actually gained from the user's response. How would you address the concern that negative user feedback (e.g., "your answer is wrong") is itself informative? Have you experimented with more fine-grained information gain measures to guide the regularization, and if so, how did they compare? A clear justification or ablation study would help validate this design choice.

2. How sensitive is BAO to the quality of the teacher model? If a weaker model were used to generate the behavior-enhanced demonstrations, would the downstream RL performance still show similar improvements? This would help disentangle the contribution of the behavioral framework from the teacher model's quality.

3. In Figure 2, the baseline with w=0 (UserRL) already shows that the agent over-engages with users. Could this be addressed simply by improving the base reward function to incorporate an interaction cost, rather than requiring the full BAO framework? It would be valuable to compare BAO against a baseline that directly integrates a well-tuned interaction cost into the reward signal during standard RL training.

**Limitations:**

yes

**Strengths And Weaknesses:**

Strengths:

1. Relevant and practical problem setting. Training proactive agents that balance task effectiveness with user effort is an important problem as LLM agents are deployed in interactive, user-facing scenarios. The paper addresses a meaningful gap in the agentic RL literature.

2. Thorough ablation studies. The ablation results in Table 1 and Figure 6 clearly demonstrate the contributions of behavior enhancement and behavior regularization.

3. The observation that behavior regularization helps mitigate reward hacking in Section 5.5 is an interesting and practically relevant finding, especially given the increasing use of LLM-as-judges in RL pipelines.

Weaknesses:

1. The multi-objective conflict is not convincingly motivated. The paper frames the core challenge as a conflict between task performance and user engagement, but the motivation for this MOO formulation has several issues:

 - The two objectives appear positively correlated, not conflicting. Fig 1(Right) visualizes a monotonically increasing relationship between user engagement and task performance. While the authors define "conflict" as the inability to simultaneously maximize performance and minimize engagement, the visual presentation suggests these are complementary rather than opposing objectives. In classical MOO literatues, a Pareto frontier typically illustrates a trade-off region; here, it more closely resembles a resource-efficiency curve (more engagement leads to more performance). The paper would benefit from clearer discussion of why this constitutes a genuine multi-objective conflict rather than a standard efficiency concern.

 - Why can't task performance alone, under a fixed budget, yield optimal user interaction behavior? Under the existing interaction budget T, optimizing task performance alone (w=0) should, in principle, incentivize the agent to balance exploration and exploitation naturally. If excessive user engagement is truly suboptimal for task performance, the agent should learn to avoid it. The authors show in Figure 2 that the w=0 agent over-uses user interactions, but this may simply mean the task reward signal does not sufficiently penalize redundant interactions — a reward design issue rather than an inherent multi-objective conflict.

 - Could this problem be formulated as a constrained optimization problem instead? The problem could alternatively be formulated as: maximize R(\tau) subject to U(\tau) \leq k, or minimize U(\tau) subject to R(\tau) \geq a threshold. The Lagrangian relaxation would recover a weighted formulation similar to Eq.(1). A more rigorous discussion of this connection is warranted.


2. Writing quality issues.

 - The abbreviation "BAO" first appears in the abstract without being spelled out as "Behavioral Agentic Optimization." Abbreviations should be introduced with their full form at first occurrence.

 - The Introduction would benefit from a concrete motivating example (e.g., a Teach Image or a simple toy-problem scenario) to illustrate the multi-objective conflict before diving into the formal problem setup. Currently, the Introduction discusses the trade-off abstractly, which makes it harder for readers to develop intuition.

 - Some notation is introduced abruptly (e.g., the structured action space A := A_u & A_e in Section 3) without sufficient motivation for why this particular decomposition is necessary or how it maps to real-world agent architectures.

---

> ### Author Rebuttal · Authors · 2026-03-30
>
> We thank the reviewer for the useful comments and provide rebuttals as follows.
>
> - W1: Clarification of MOO formulation
>
>     Our goal is to claim that task performance and user engagement correspond to two fundamentally different objectives: task performance measures solution quality, while user engagement reflects user burden. Relying on excessive user engagement may improve performance, but it also increases user cost, making these objectives inherently competing.
>
>     - “Positive correlation vs conflict.”
>
>     In Fig 1, the monotonic trend shows that higher performance can be obtained by spending more interaction budget, which is precisely the source of conflict. The agent should maximize performance while **minimizing** unnecessary user interaction instead of maximizing both performance and interaction simultaneously. Therefore, the desirable region is in upper-left, and the Pareto frontier captures the best achievable trade-off between these two goals, and we aim at minimizing the "infeasible region" in the revised Fig 1 (right) (https://anonymous.4open.science/r/BAO/Figure%201%20revised.png)
>
>     - “Why can't task performance alone, under a fixed budget, yield optimal user interaction behavior?...”
>
>     From a problem formulation perspective, a single hard constraint on user-involved actions is insufficient, as the appropriate level of interaction varies across tasks and user preferences. A multi-objective formulation better captures this.
>     Moreover, optimizing only for task reward under a budget constraint or with interaction penalties is not sufficient to train proactive agents, as shown in Fig. 2 and experiments in Q3: it fails to simultaneously improve task performance and reduce user engagement. Explicitly shaping multi-turn behaviors, such as retrospective reasoning and prospective planning, is therefore necessary for efficient interaction. This is further supported by the ablation studies in Sec. 5.3, which show that reward shaping alone is insufficient.
>     We have clarified this in the revision.
>
>     - “Constrained formulation”
>
>     Yes, it can also be written as constrained optimization if we have a **fixed** user-envolved interaction budget k. Our MOO formulation is closely related to the corresponding Lagrangian relaxation. We adopt the multi-objective view because it provides a convenient way to study and control the trade-off over task performance and user engagement with a **flexible** interaction budget with diverse task difficulty and user preference. We have added this connection explicitly in the revision.
>
> - W2: Writting
>
>      For the abbreviation issue, we have corrected it in the revision
>
>      For the illustration, we have revised Figure 1 (right) to better present the Pareto frontier.
>
>      Regarding the notation, we would like to clarify that it is defined in the submission (Lines 102–109). This decomposition is necessary and is central to our formulation and directly used in behavior regularization (Sec. 4.2.2) and evaluation metrics (Sec. 5.1). We also supplement the definition of them in the experiment tasks in Appendix B.1. We have improved the presentation to make this motivation more immediately clear.
>
> - Q1: Information-seeking regularization
>
>      We agree that negative feedback provides information. However, repeated user feedback often yields low marginal information gain and increases unnecessary user efforts, especially when the agent produces semantically similar answers without new reasoning or environment interaction.
>
>     We further evaluate information diversity via BLEU scores and semantic embedding similarity (Sec. 5.4), where BAO shows improved information gathering. We agree that incorporating these measures into training is an interesting direction for future work.
>
> - Q2: Sensitivity to teacher model
>
>     We replace GPT-4o with a weaker teacher (GPT-4o-mini) on Telepathy-Gym (Qwen3-1.7B as base models):
>
>     ||Pass@A-1($\uparrow$)|Pass@A-2($\uparrow$)|Score($\uparrow$)|AR($\downarrow$)|
>     |-|-|-|-|-|
>     |UserRL|0.1707|0.2439|0.4634|0.7619|
>     |BAO|0.5122|0.5366|0.5366|0.1423|
>
>      BAO maintains substantial improvements over the UserRL baseline under the same weaker teacher model.
>
> - Q3: Comparison to reward-penalty baseline
>
>     We adopt the intrinsic reward for penalty in CollabLLM [1] and upgrade it from DPO to GRPO, and name this baseline as CollabLLM-GRPO. The evaluations on FunctionGym with Qwen3-4B as base models are:
>     ||Score($\uparrow$)|AR($\downarrow$)|
>     |-|-|-|
>     |UserRL|0.5256|0.3758|
>     |CollabLLM-GRPO|0.4872|0.2917|
>     |BAO|0.6923|0.2148|
>
>     We can see that simply penalizing interaction reduces AR but also degrades performance, leading to overly passive policies. In contrast, BAO improves both metrics, indicating that structured behavior learning is key to improving the Pareto frontier.
>
> [1] Shirley Wu, et al. Collabllm: From passive responders to active collaborators. ICML 2025

---

> > ### Author Rebuttal · Reviewer_oZ1v · 2026-04-02
> >
> > I thank the authors for the thorough rebuttal. The additional experiments are convincing, and Q1–Q3 are well addressed.
> >
> > I still have a minor reservation about the "Pareto frontier" framing. By standard definition (see e.g. wiki: https://en.wikipedia.org/wiki/File:Front_pareto.svg, or other classical literatures on MOO), a Pareto frontier is the set of non-dominated solutions where improving one objective necessarily degrades another. Since BAO simultaneously improves task performance and reduces user engagement over the UserRL frontier, the relationship shown is closer to an efficiency improvement than a genuine multi-objective trade-off. I would suggest the authors use the Pareto / MOO terminology more carefully in the final version, or add a brief discussion clarifying the distinction.
> >
> > Overall I think this is a borderline paper, but I am willing to raise my score, given the strong empirical results and the improvements made during rebuttal,.

---

> > > ### Author Response · Authors · 2026-04-06
> > >
> > > We thank the reviewer for the time and effort throughout the review process, and we sincerely appreciate the helpful feedback and comments on our paper. We will incorporate the points from the rebuttal into the revised manuscript.

---

### Decision · Program_Chairs · 2026-04-30

**Decision:**

Accept (regular)

**Comment:**

This paper tackles the practical trade-off between task performance and user engagement in proactive LLM agents, proposing BAO—a two-stage framework combining behavior-enhanced SFT (retrospective reasoning and prospective planning) with behavior-regularized RL. Experiments on UserRL benchmarks show consistent improvements over baselines across model scales, with valuable ablations and an insightful reward-hacking analysis. The rebuttal effectively addressed most concerns through additional experiments (teacher sensitivity, CollabLLM-GRPO comparison, UserBench generalization), and all reviewers converged on acceptance. However, the "Pareto frontier" framing is imprecise: BAO strictly dominates baselines on both objectives rather than navigating a genuine trade-off. Authors should refine this terminology, complete UserBench metrics (AR, Pass@A-k), and extend hyperparameter sensitivity analysis to Turtle-Gym in the camera-ready.